# The maluma/takete effect is late: No longitudinal evidence for shape sound symbolism in the first year

David M. Sidhu[1]*, Angeliki Athanasopoulou[2], Stephanie L. Archer[3], Natalia Czarnecki[4], Suzanne Curtin[5], Penny M. Pexman[4]

1 Department of Psychology, Carleton University, Ottawa, Canada, 2 School of Languages, Linguistics, Literatures, and Cultures, University of Calgary, Calgary, Canada, 3 Department of Linguistics, University of Alberta, Edmonton, Canada, 4 Department of Psychology, University of Calgary, Calgary, Canada, 5 Department of Child and Youth Studies, Brock University, St. Catharines, Canada

* david.sidhu@carleton.ca

**Data Availability Statement:** All data and analysis files are available from the OSF database (https://osf.io/sd62f/).

## Abstract

The maluma/takete effect refers to an association between certain language sounds (e.g., /m/ and /o/) and round shapes, and other language sounds (e.g., /t/ and /i/) and spiky shapes. This is an example of *sound symbolism* and stands in opposition to arbitrariness of language. It is still unknown when sensitivity to sound symbolism emerges. In the present series of studies, we first confirmed that the classic maluma/takete effect would be observed in adults using our novel 3-D object stimuli (Experiments 1a and 1b). We then conducted the first longitudinal test of the maluma/takete effect, testing infants at 4-, 8- and 12-months of age (Experiment 2). Sensitivity to sound symbolism was measured with a looking time preference task, in which infants were shown images of a round and a spiky 3-D object while hearing either a round- or spiky-sounding nonword. We did not detect a significant difference in looking time based on nonword type. We also collected a series of individual difference measures including measures of vocabulary, movement ability and babbling. Analyses of these measures revealed that 12-month olds who babbled more showed a greater sensitivity to sound symbolism. Finally, in Experiment 3, we had parents take home round or spiky 3-D printed objects, to present to 7- to 8-month-old infants paired with either congruent or incongruent nonwords. This language experience had no effect on subsequent measures of sound symbolism sensitivity. Taken together these studies demonstrate that sound symbolism is elusive in the first year, and shed light on the mechanisms that may contribute to its eventual emergence.

## Introduction

For the most part, the sound of a word is arbitrarily associated with its meaning. As an example, one must learn that the set of speech sounds in *cat* refers to a feline with whiskers and claws. The speech sounds themselves do nothing to help in this process. This principle of arbitrariness was once thought to be more or less ubiquitous in language (e.g., [1]). However, there

**Funding:** This research was supported by the Social Sciences and Humanities Research Council of Canada (SSHRC; https://www.sshrc-crsh.gc.c) through an Insight Development Grant (430-2017-00003) to PP and SC. The funders had no role in study design, data collection and analysis, decision to publish, or preparation of the manuscript.

**Competing interests:** The authors have declared that no competing interests exist.

is growing appreciation for the role of *iconicity* in language: instances in which the form of language imitates or depicts meaning in some way (see [2–4]). While iconicity can emerge in tone of voice or gesture, here we focus on instances in which speech sounds imitate or depict meanings. For instance, the speech sounds in *meow* do more than serve as a set of sounds to associate with a meaning (as with *cat*). They also imitate the sound to which they refer, making *meow* an iconic word.

Importantly, iconicity can emerge beyond the imitation of sounds. Speech sounds can also imitate non-sound features through analogy (though for other potential mechanisms see [5]). For example, the speech sounds in *cactus* are abrupt, with sudden changes in amplitude [6]. In this way, *cactus* imitates the visual property of spikiness through analogy. The term for this is *sound symbolism*: associations between speech sounds and perceptual and/or semantic properties. In this paper we focus on the most well-studied instance of sound symbolism: associations between speech sounds and either round or spiky shapes (the *maluma/takete effect*; [7]). For instance, when adults are shown shapes like those in Fig 1 and asked which is a *maluma* and which is a *takete*, roughly 90% of individuals will respond that the round shape is *maluma* while the spiky shape is *takete* (though see [8] as well as [9] for variation based on language differences). In general, sonorants (e.g., /l/), voiced stops (e.g., /b/), unvoiced fricatives (e.g., /s/), and rounded-back vowels (e.g., /u/) tend to be associated with round shapes [10, 11]. Conversely, voiceless stops (e.g., /k/), affricates (e.g., /tʃ/) and front-unrounded vowels (e.g., /i/) tend to be associated with spiky shapes [10]. These associations emerge on forced choice tasks in a bias to pair nonwords with certain shapes (e.g., [12]), on implicit tasks in facilitated responses to congruent nonword-shape pairings (e.g., [13]), and in different neural responses to congruent vs. incongruent pairings (e.g., [14]). In this study we address a pair of unanswered questions. When do children become sensitive to shape sound symbolism? And what developmental milestones are related to that sensitivity?

These questions are theoretically important for several reasons. For one, some have proposed that sound symbolism and iconicity play a role in language acquisition (e.g., [15]; for a review see [16]). There is now a wealth of evidence that iconic words are easier for children to acquire ([17, 18]; termed *local enhancement* by [16]). It could be that the imitative links in iconic words help children associate a sound with a meaning. Imai et al. [19] also found evidence that infants had an easier time generalizing an iconic label to a novel instance of the word's referent (e.g., learning that a certain kind of movement is called *batobato* whether it is performed by a rabbit or a bear). Imai and Kita [15] speculated that the existence of iconic words could help children realize that sounds refer to things in the world (i.e., gain referential insight). In their sound symbolism bootstrapping hypothesis, Imai and Kita [15] claim that sound symbolism provides infants with the insight that speech sounds refer to things in the world. For example, 14-month-olds match novel objects with sound symbolic words (kipi,

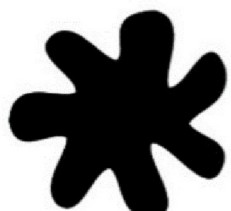 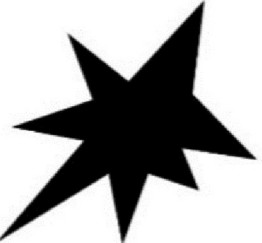

**Fig 1. Examples of shapes used in studies of the maluma/takete effect.**

moma) after habituation to the sound symbolic pairings [20]. However, if that is the case, then we should expect infants to demonstrate this type of insight much earlier. Yet there has not been evidence for this *general enhancement* (reviewed in [16]). Knowing *when* infants become sensitive to sound symbolism can help understand its contribution to word learning. For example, it would be informative to know whether sensitivity develops before or *after* infants understand that sounds refer to things.

Knowing when sensitivity to sound symbolism develops with regards to developmental milestones and experiences would also be informative for theories of sound symbolism. It is still unclear which linguistic aspects play a primary role in the maluma/takete effect. It could be that the phonetic properties of speech sounds play a key role (e.g., that /t/ is associated with spiky things because of the sudden changes in amplitude). One could also make a case for experiences of articulation (e.g., that /t/ is associated with spiky things because of the abrupt release of the articulator after the closure). One might also suggest that the shapes of letters (e.g., the round "o") or the lips (e.g., the lips for rounded vowels) play important roles in the effect (for a discussion of these different possibilities see [9, 21–24]). Of course, none of these possibilities are mutually exclusive, and all may play a role to different degrees (perhaps changing in relative importance throughout development). Knowing if sensitivity to sound symbolism emerges before or after experience with these different linguistic aspects could thus be informative.

In addition, it is not yet known *how* speech sounds get associated with shapes. For instance, some have suggested that sound symbolic associations arise from an internalization of statistical occurrences in the environment [6, 25, 26]. Others have proposed that the maluma/takete effect could stem from knowledge of patterns in language, in which certain sounds occur in words for certain shapes [27, 28]. Testing whether sensitivity to sound symbolism emerges before or after infants begin exploring the environment or acquiring vocabulary would shed light on these possibilities.

### Previous work on infant sound symbolism

There have been previous attempts to investigate whether infants of different ages are sensitive to sound symbolism. Maurer et al. [29] gave two and a half year olds the sort of task illustrated in Fig 1, and found that infants' responses were comparable with those of adults, thus suggesting that sensitivity emerges by 2.5 years of age. In order to test younger preverbal infants, researchers have instead used techniques that do not require participants to understand instructions or make verbal responses. The most common approach has been a looking time preference task. Infants sit on a caregiver's lap and have their attention directed towards a screen showing either one shape or two (one round and one spiky). While looking at the screen, infants hear a round- or a spiky-associated nonword. In the one-shape version, the dependent measure is the amount of time infants look at the screen for a congruent vs. incongruent nonword. In the two-shape version, the dependent measure is looking time to the congruent vs. incongruent shape.

Ozturk et al. [30] used a one-shape looking time preference task with four-month-olds. They found that infants looked longer at the incongruent pairings as compared to the congruent pairings, suggesting very early sensitivity to sound symbolism. However, this experiment only included 12 infants. In addition, two follow up experiments that presented nonwords contrasting only in either vowels or consonants did not find a significant effect. Pejovic and Molnar [31] attempted to replicate this effect with a sample of 26 four-month-olds using the same task, and found no differences between congruent and incongruent pairings. However, they did find a significant difference in a sample of 12-month-olds (n = 26).

There is also neuroimaging evidence of sensitivity to sound symbolism emerging by 12 months. Asano et al. [14] studied 49 11-month-olds with electroencephalogram (*EEG*) based measures of brain activity. Participants were first shown a shape and then heard a congruent or incongruent nonword. The researchers found differences between congruent and incongruent pairings in phase synchronization, and in the intensity of infants' N400 responses (a negative going waveform sensitive to semantic integration). This finding was complimented by that of a study of 22 11-month-olds using near infrared spectroscopy (NIRS) to measure brain activity [32]. This study found greater activity in the right superior temporal sulcus for congruent pairings of *round* nonwords. Importantly, this area has been implicated in the processing of iconicity in adults [33, 34].

The literature on infant sound symbolism was recently summarized in a meta-analysis by Fort et al. [35]. They combined a total of 44 effect sizes, from 22 different studies, which included infants ranging in age from 4–38 months. The studies included both behavioural (e.g., forced choice and looking time) and neural measures. Fort et al. found that the effect of sound symbolism (i.e., comparing responses to congruent vs. incongruent pairings) was significantly different than zero. However, after adding nonword type (i.e., round or spiky), age, their interaction, and publication status, the effect of sound symbolism was no longer significant. They did, however, find a larger effect for round vs. spiky nonwords. When analyzing each nonword type separately, they found a marginal effect of sound symbolism for round nonwords ($p = .062$), but not spiky nonwords. Notably, age was a significant predictor in the case of spiky nonwords, suggesting that sensitivity to spiky sound symbolism increases with age.

In sum, the literature on infant sound symbolism is equivocal. Many studies have included small samples: the 22 published studies examined by Fort et al. [35] included an average of just over 19 participants. Overall, after including control variables, there was not a significant effect of sound symbolism in Fort et al.'s meta-analysis. However, some of the studies reviewed above suggest that there is an emerging sensitivity to sound symbolism by about 12 months of age. In the present study we conducted a thorough test of infants' emerging sensitivity to sound symbolism, using a longitudinal design to investigate whether that sensitivity emerges in the first year of life and what milestone or experience might be associated with its emergence.

## The present study

In the present study we first verified that the maluma/takete effect will be observed in adults using our stimuli (Experiments 1a and 1b). Then we conducted a longitudinal experiment testing sound symbolism at 4, 8 and 12 months (Experiment 2). We measured several individual differences at each time point to gain insight into the developmental milestones and experiences that are related to sound symbolism. We collected measures of infants' babbling to quantify their experience articulating speech sounds. We also included measures of their ability to move through the environment, to quantify their tactile experience of the world, and their knowledge of the statistical regularities in the world. Finally, we measured infants' receptive and productive vocabulary, to quantify their overall language development, and to estimate their knowledge of patterns in language. Finally, in Experiment 3, we conducted a study in which we directly manipulated the statistical regularities in the environment. Caregivers either produced congruent or incongruent labels for a round or a spiky shape at home. We then examined the effect of this experience on infants' sound symbolic associations.

## Experiment 1a

### Methods

**Ethics approval.**   All experiments were approved by the Conjoint Faculties Research Ethics Board at the University of Calgary.

**Participants.**   Participants were 30 undergraduates at the University of Calgary who participated in exchange for course credit. All participants provided written informed consent. Records of demographics for these participants were not retained. All participants reported English fluency and normal or corrected to normal vision.

**Stimuli.**   We used the same shape stimuli in each of the experiments reported here. Because these shapes needed to be taken home by families in Experiment 3, we created 3D-printed models of a round and a spiky shape. Wireframes of these stimuli are available at: https://osf.io/sd62f/. These were painted either green or orange. The visual stimuli in Experiments 1 and 2 were photographs of these shapes, see Fig 2, also available at: https://osf.io/sd62f/. On each trial in Experiment 1a participants saw a round and a spiky shape of the same colour. There were two different versions of each same colour pair, with the objects

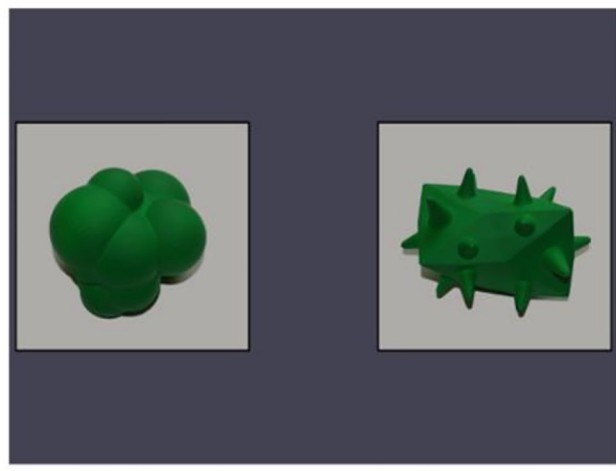

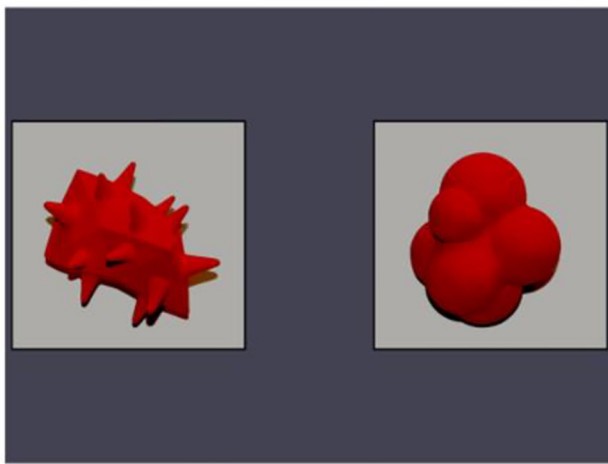

Fig 2. **Examples of round and spiky object stimuli.**

photographed from different angles. This created eight shape pairs. We collected pilot ratings on these images from a separate group of adult participants (n = 12) to ensure that the shapes paired in each round/spiky image pair were similar in rated visual interest.

As in previous studies of infant sound symbolism (e.g., [30, 31]), nonword auditory stimuli consisted of four disyllabic CVCV nonwords. Two of these (i.e., [boʊboʊ] and [lulu]) contained round-associated phonemes (voiced stops or sonorant consonants; and rounded back vowels), and two (i.e., [keɪkeɪ] and [tʃitʃi]) contained spiky-associated phonemes (voiceless stops or affricate consonants; and unrounded front vowels; [10]). In addition, for the purpose of our subsequent experiments with children, one of each type of nonword contained consonants that are relatively more early-acquired, in that children typically produce them around three years of age (i.e., /b/ and /k/), and one of each contained consonants that are relatively later-acquired, in that children typically produce them around 4.5 years of age (i.e., /l/ and /tʃ/; [36]). Nonwords were recorded by a Canadian-English female native speaker in infant directed speech.

**Procedure.** Participants took part in person in our laboratory at the University of Calgary. The experiment was run using E-Prime 2.0 software. Participants wore sound attenuating headphones. On each trial, participants heard a nonword and this was repeated until an eventual response was made (to match the stimulus repetition in the planned infant experiment). Participants were also presented with a shape pair as the audio began, and were asked to decide which shape best fit the nonword. Responses were made via button box. Stimulus order, and the pairing between nonwords and shape pairs, was randomized across participants.

## Results

All mixed effect models were computed using the "lme4" [37] and "afex" [38] packages in the statistical software R [39]. Here and in subsequent experiments, we included random intercepts for subject and item. We also included a random subject slope for predictor(s) of interest. To deal with convergence issues and singular fits we attempted these remedies, in the following order: increasing the number of iterations, switching to the "bobyqa" optimizer, removing the correlation between the random subject intercept and slope, removing the random subject slope, and removing the random intercept with the lowest associated variance (see [40, 41]). Data and analysis code for all experiments can be found here: https://osf.io/sd62f/.

The present analysis began as a logistic mixed effects regression. The dependent variable was whether or not a participant chose the round shape on a given trial. The predictor was nonword type (using dummy coding; spiky [0] vs. round [+1]). However, there was no variance in the random effects, and so we proceeded with a simple logistic regression. Note that the results are identical if the random subject and item intercepts showing zero variance are included. The analysis revealed a significant effect of nonword type such that participants were 6.68 times more likely to choose the round shape for a round vs. a spiky nonword ($b = 1.90$, $p < .001$). See Fig 3. Participants chose the sound symbolically congruent shape on 70.83% of trials (95% CI = [62.95, 78.71]). We conducted a follow up analysis in which the dependent variable was whether or not a participant chose the *congruent* shape on a given trial. The predictor was nonword type. This analysis suggested that congruent choices were made significantly more often on round (81.67%, 95% CI [74.76, 88.57]) vs spiky (60%, 95% CI [48.61, 71.39]) trials ($b = 1.23$, $p < .001$). Tables for this (and all other) supplementary analyses can be found in the Online Supplementary Materials; see S1 Table in S1 File. See Table 1 for results by item.

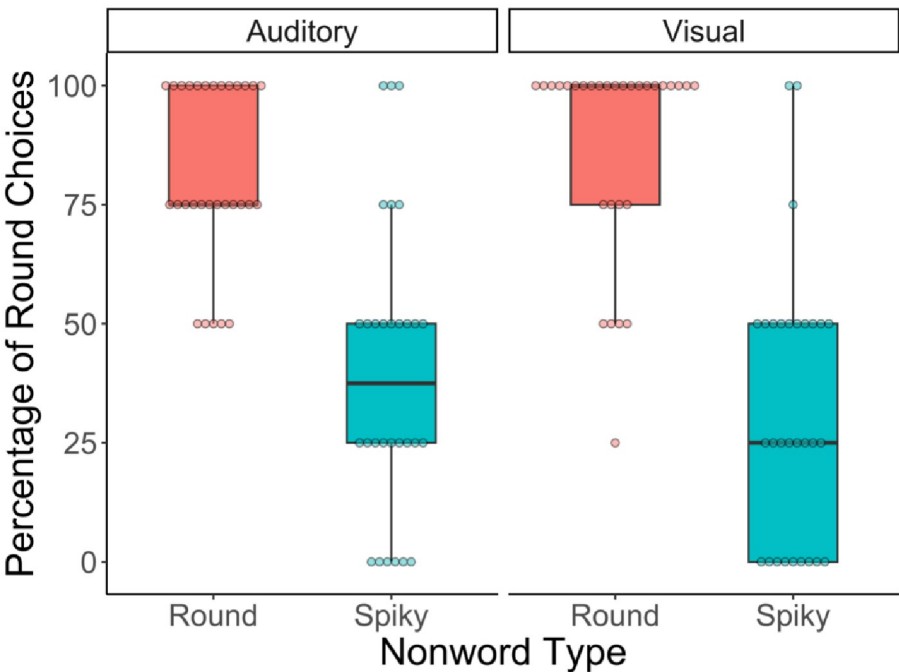

**Fig 3. Proportion of trials on which adult participants chose the round shape of the pair in Experiments 1a and 1b.** This plot shows the percentage of trials on which each adult participants chose the round shape as the best match for either type of nonword, in either the auditory (Experiment 1a) or visual (Experiment 1b) experiment. Whiskers represent maximum and minimum values, boxes represent the first quartile, median, and third quartile. Dots represent individual subject means. Ns = 30.

## Experiment 1b

### Method

**Participants.** Participants were 30 individuals (24 female; $M_{Age}$ = 26.87, $SD_{Age}$ = 6.29) recruited through the online platform Prolific (https://www.prolific.co/). Participants were paid at a rate of 7.50 GBP per hour. All participants provided electronic informed consent. All participants reported English fluency and normal or corrected to normal vision.

**Stimuli.** The stimuli were the same as in Experiment 1a.

**Procedure.** The procedure was the same as in Experiment 1a, except that here nonwords were only presented visually onscreen in their written forms (i.e., "boeboe", "looloo", "kaykay"

**Table 1. The percent of trials on which participants paired nonwords with congruent shapes in Experiments 1a and 1b.**

| Nonword | Experiment 1a | Experiment 1b |
|---|---|---|
| | M [95% CI] | M [95% CI] |
| [boʊboʊ]—"boeboe" | 81.67% [71.29, 92.05] | 86.67% [75.78, 97.56] |
| [lulu]—"looloo" | 81.67% [68.25, 95.08] | 88.33% [77.72, 98.94] |
| [keɪkeɪ]—"kaykay" | 61.67% [46.41, 76.92] | 81.67% [69.18, 94.15] |
| [tʃitʃi]—"cheechee" | 58.33% [43.56, 73.11] | 53.33% [37.12, 69.55] |

Nonwords are shown in their phonological form, and in their written form (presented in Experiment 1b).

and "cheechee") instead of auditorily. The nonword appeared below the shape pair image. The experiment was run in PsychoPy, hosted by the by the platform Pavlovia (https://pavlovia.org/).

## Results

This analysis was the same as that in Experiment 1a. This time a model with random subject and item intercepts, as well as a random subject slope for the effect of nonword type, was able to be fit. The mixed effects logistic regression revealed a significant effect of nonword type such that participants were 57.75 times more likely to choose the round shape for a round vs. a spiky nonword ($b$ = 4.06, $p <$ .001). See Fig 3. Participants chose the sound symbolically congruent shape on 77.50% of trials (95% CI = [69.90, 85.10]). A follow up analysis suggested that congruent choices were made significantly more often for round (88%) vs spiky (68%) nonwords ($b$ = 2.11, $p <$ .001), see S2 Table in S1 File. This difference was driven by results for the nonword *cheechee* which was only paired with a congruent shape 53% of the time (*boeboe* = 87%; *looloo* = 88%; *kaykay* = 82%), see Table 1. This rate of pairing for *cheechee* was lower than in Experiment 1a (58%) and could be due to its rounded orthography (see [42]), a factor that will not play a role with infants in Experiments 2 and 3.

## Discussion

We observed a maluma/takete effect with adult participants for our nonwords and visual stimuli. This is notable because the majority of sound symbolism studies have used simple two-dimensional shapes as stimuli (as illustrated in Fig 1; cf [43]), rather than the images of three-dimensional objects presented here.

## Experiment 2

### Method

**Participants.** Based on prior work using preferential looking with samples ranging from 30–50 participants (e.g., [44–48]) and accounting for possible attrition given the longitudinal design, we recruited 63 infants (31 females) who were between the ages of four and five months ($M$ = 4.87 months; $SD$ = 0.54) at the time of their first testing session. They were tested again at 7–8 and 11–12 months. Infants were recruited from the University of Calgary's Child and Infant Learning and Development (Ch.I.L.D.) research group database. Infants' parents or guardians provided written informed consent. The infants received a t-shirt, a bath toy, and a certificate for participating in the experiment. All infants were exposed to English at least 50% of the time (3 who did not meet this criterion were excluded from analyses); 28 were exposed to another language (Cantonese, Spanish, Afrikaans, German, French, Russian, Hungarian, Dutch, Tagalog, Urdu, ASL, Swahili, Italian, German, Mandarin). Parents reported infants' ethnic background: 2 Asian; 42 White; 1 Black; 1 South Asian; 1 Arabic; 12 Mixed Ethnicity. Most parents (68%) had a university degree or higher. One infant was preterm, and their age was corrected to full term. There were two pairs of twin siblings.

**Stimuli.** The experimental stimuli consisted of the same visual and auditory nonword stimuli as in Experiment 1a. The materials for this experiment included a Developmental Questionnaire with 40 questions about children's physical, language, and communicative abilities that caregivers were asked to fill out weekly throughout the study in Qualtrics (Qualtrics, Provo, UT). A copy of this survey can be found here: https://osf.io/djgfx. This was included for the purposes of exploratory analyses to investigate if any developmental milestones predicted sensitivity to sound symbolism.

There were three components to the questionnaire. The first component was a set of questions about children's physical milestones (e.g., standing, walking, etc.). The caregivers were asked to indicate their child's mastery of each milestone (9 in total) using a six point scale ("not yet", "just starting", "gaining confidence", "showing confidence", "doing moderately well", "mastered"). The second component was a set of questions about babbling of specific speech sounds. We included the speech sounds used in our nonwords and also speech sounds that are typically acquired earlier or are frequent in English: /p, t, k, b, d, g, m, n, , f, s, v, z, w, j, l, ɹ, ʧ, i, u, eɪ, oʊ, ɪ, ʊ, ɛ, æ, ɑ, aɪ, aʊ/. For each speech sound, the caregiver was asked to indicate the extent to which their child was producing the sound when they were babbling on a five point scale ("not yet", "maybe", "probably yes", "definitely yes", "frequently"). Parents were also asked about the extent to which their child was engaging in vocal play (e.g., experimenting with producing speech sounds before producing proper phonemes). The third component was a set of questions about vocabulary development. Here the caregivers were asked to indicate for each of the 10 words that children commonly develop first (mom, dad, baba, yumyum, uhoh, bye, hi, grrr, woof, meow) whether their child understands the word on a five point scale ("not yet", "maybe", "probably yes", "definitely yes", "frequently").

When the child reached the age of 7–8 months (i.e., after their second lab visit), the set of questions for vocabulary slightly changed in that the caregivers were only asked to report any protowords the children might be producing and whether and how often they use Baby Sign with their child. The caregivers reported vocabulary development with the MacArthur-Bates Communicative Development Inventories Short-Form Level 1 [49] at the second and third visit (see below), replacing the 3 sets of questions described above. We also included another set of questions about the children's communicative development. From 7–8 months onwards the parents were asked to indicate whether children show an understanding of the communicative intent of others (e.g., responds to their name), whether they express communicative intent (e.g., extends arm to show what they hold, waves "bye-bye", etc.), and whether they were using gestures to communicate (pointing, clapping, or other), on a three point scale ("not yet", "sometimes", "often").

**Procedure.** Infants were brought into the lab by their caregiver three times during the course of the experiment: at 4–5 months for their first visit, at 7–8 months for their second visit, and at 11–12 months for their third visit.

At the start of the first visit, while the infant familiarized themselves with their new surroundings, the caregiver and research assistant went over the consent process to confirm participation, and then filled out a demographics questionnaire and a language questionnaire to determine the nature of the infant's language environment. Once this was completed, the caregiver and infant were seated in the testing room (sound attenuated) to begin the study.

At each visit, to test infant sensitivity to sound symbolism, we used the Preferential Looking paradigm [e.g., 50]. Infants were seated on their caregiver's lap or in a highchair with their caregiver sitting next to them. Infants faced a large smart-board screen which was 150 cm away from the infant where the visual stimuli were shown. The screen was 127 cm diagonal (thus subtending a visual angle of 45.89˚). Caregivers were given noise cancelling headphones with music playing to prevent any influence on the infants' looking patterns. The study was run using the Habit X 1.0 software (see [51]), and the infants' looks were recorded using a camera (Canon VIXIA HF R600) positioned discreetly underneath the screen. Before every trial (and before the pre-test), a colorful bouncing ball appeared on the screen to attract the infant's attention. The duration of the bouncing ball was controlled by the experimenter and stayed on the screen until the infant turned their attention to the screen (typically 3–4 s). Next, we presented infants with a 20-second pre-test (spinning pinwheel with music) to further capture their attention to the screen. Following the pre-test, the test trials were presented, where

infants saw the visual stimuli (pair of objects) accompanied with the auditory nonword (a single label repeated 12 times), which was kept at a range of 65 to 70dB. These recordings are available at: https://osf.io/sd62f/. The entire trial lasted 15 seconds (ISI 151ms), unless the child did not attend for two seconds in which case it was terminated. Each infant was presented with 8 of these randomized trials, hearing each nonword twice.

We set a minimum look time of 2 seconds so that the infant had the opportunity to hear at least one token of the word. On the rare occasion that an infant did not attend for a minimum of two seconds, the bouncing ball and pre-test were repeated to recapture the infant's attention. After the test trials, we presented a post-test, which was identical to the pre-test (i.e., a spinning pinwheel with music for 20 seconds).

After the first visit, caregivers were sent weekly the Development Questionnaire using Qualtrics (Qualtrics, Provo, UT) until they were scheduled to come in for their second visit at 7–8 months-of-age.

During the second visit, parents filled out the MacArthur-Bates Communicative Development Inventories Short-Form Level 1 [49] to gain more information on the infants' language understanding. Following this, infants were again presented with the preferential looking task. After the visit and weekly thereafter families were sent the Development Questionnaire, until they came in for their final visit when their infant reached the age of 11–12 months.

The final visit was identical to the second visit. Parents once again filled out the MacArthur-Bates Communicative Development Inventories Short-Form Level 1 [49], to track language development, and the infants were presented with the preferential looking task. A total of 48 infants completed all three visits.

## Results

For each trial, we first calculated the amount of time spent looking at one of the two shapes onscreen (i.e., *looking time*), removing any time spent not looking at either shape from the analyses. We excluded trials on which a participant did not spend at least two seconds looking at either shape (1.56% of trials). We then calculated the percentage of looking time that was spent looking at the round object. Because participants could only look at the round or the spiky object, looking time to the round object was redundant with the percentage of looking time spent looking at the spiky object. Thus, 50% is the chance level, looking time above 50% indicates that infants are looking more to the round than the spiky object, and looking time below 50% indicates that infants are looking more to the spiky than the round object.

Analyses consisted of linear mixed effects regressions. Our dependent variable was percent of looking time to the round object. We centered this variable by subtracting 0.5 in order to allow interpretation of the intercept. In every model, we fit fixed effects corresponding to the experimental manipulations of interest [41]. Our first set of analyses included as predictors: nonword type (using effects coding; round [+.5] vs. spiky [-.5]), visit number (using subsequent coding; i.e., comparing each visit to the one that came before), and their interaction. Table 2 summarizes the results. The intercept was significantly less than 0 ($b$ = -0.07, $p$ < .001), indicating that participants spent longer looking at the spiky shape overall. Despite matching round and spiky image pairs on rated visual interest, it may have been that the spiky images were more interesting. (Indeed, a follow up version of the task that did not include audio stimuli also found that participants looked longer at the spiky shape ($p$ = .04). This is reported in Online Supplementary Material). No significant main effect of nonword type was observed ($p$ = .43), which means that the looking time to the round object when a round nonword was played was not statistically different from when a spiky nonword was played. See Fig 4. There was also a not a significant main effect of visit three (vs. visit two $p$ = .71), indicating that the

**Table 2. Linear mixed effects regression model predicting looking time for all infants with nonword type and categorical visit.**

| Fixed Effect | B | SE | t | p |
|---|---|---|---|---|
| Intercept | -0.07 | 0.01 | 12.33 | < .001*** |
| Nonword Type | 0.01 | 0.01 | 0.78 | 0.43 |
| Visit (2 vs 1) | 0.03 | 0.01 | 2.15 | 0.03* |
| Visit (3 vs 2) | 0.01 | 0.01 | 0.37 | 0.71 |
| Nonword Type x Visit (2 vs 1) | 0.01 | 0.03 | 0.19 | 0.85 |
| Nonword Type x Visit (3 vs 2) | 0.01 | 0.03 | 0.97 | 0.34 |
| Random Effect | $s^2$ | | | |
| Participant Intercept | 0.0003 | | | |

looking time to the round object in visit three was not statistically different from that in visit two. There was however a significant main effect of visit two (vs. one; $b = 0.03$, $p = .03$), such that infants at visit two spent a greater percentage of time looking at the round shape than they did in visit one. Finally, no significant interaction between nonword type and visit was found (all $p$'s > .33). Table 2 summarizes the results. We also ran a supplementary analysis to examine if phoneme type (using effects coding; early [-.5] vs. late [+.5]) interacted with any of these predictors: it did not (all $p$'s > .39), see S3 Table in S1 File.

In order to quantify the evidence in favour of the null hypotheses, we conducted a version of the analyses using Bayesian mixed effects regression. For a detailed introduction to this approach we refer the reader to Vasishth et al. [52]. We conducted a region of practical

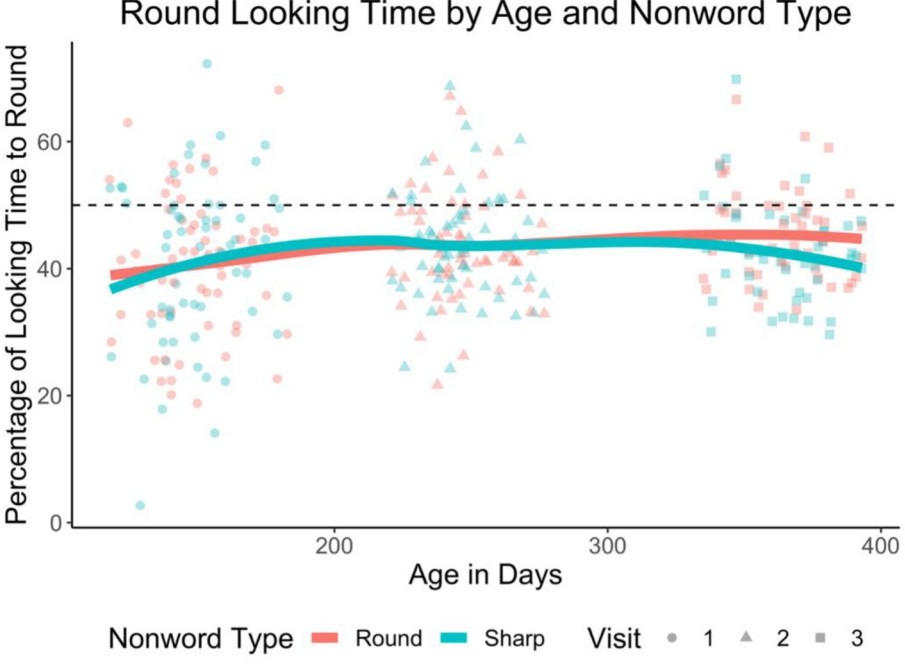

**Fig 4. Percent round object looking time by age and nonword type in Experiment 2.** This plot shows the percentage of total looking time spent looking at the round object, based on the age of the infant in days, and the type of nonword presented on each trial. The shape of each point corresponds to the visit number. Lines are LOESS functions. The dashed line corresponds to equal looking time between round and spiky shapes. $N_{Visit\ 1} = 62$, $N_{Visit\ 2} = 57$, $N_{Visit\ 3} = 49$.

equivalence (*ROPE*) analysis (see [53]). This method examines the percentage of the highest density interval (*HDI*; i.e., the interval of credible values for each parameter) that falls within a *region of practical equivalence*. That is, instead of treating the null hypothesis as a point value of 0, it treats a range of values around 0, that are practically equivalent to 0, as representing the null. Here we adopt Kruschke's (2018) suggestion of a range of +/− 0.1 standard deviations in the dependent variable as the ROPE. This equates to what Cohen [54] considered a negligible effect size. In other words, this analysis asks: when taking a range of values that we are 95% sure contain the true value of a coefficient, what percentage are practically equivalent to 0. We used the packages "rstanarm" [2.21.1] [55] and "bayestestR" [0.7.5] [56] to conduct Bayesian analyses. Models were run with 2000 iterations, the first 1000 of which were treated as a warm up. Four chains were run. If these models did not show good convergence (i.e., an effective sample size of at least 10% that of the total number of samples; [52]) additional iterations were added. Fixed effects were given a generic normal prior centered at 0, with a standard deviation of 2.5. When examining the interaction between nonword type and visit, only 36.93% (Visit 3 vs 2) and 55.05% (Visit 2 vs 1) of each HDI fell within the ROPE. However, 82.62% of the HDI for nonword type fell within the ROPE.

We next ran a set of analyses that treated age as a continuous variable (after being standardized). Table 3 summarizes the results. We again found no significant main effect of nonword type ($p$ = .49) and no significant interaction between nonword type and age ($p$ = .30). There was however a significant main effect of age ($b$ = 0.01, $p$ = .037) such that older children spent a greater percentage of time looking at the round shape than did the younger children. A Bayesian analysis revealed that 79.40% of the HDI for the interaction between nonword type and age fell in the ROPE, as did 86.72% of the HDI for nonword type. We again ran a supplementary analysis examining if phoneme type interacted with any of these predictors: it did not (all $p$'s > .35), see S4 Table in S1 File. Note that here and in the analysis with age as a categorical predictor, the pattern of results do not change when only including children who attended all three sessions ($n$ = 48). See S5 and S6 Tables in S1 File. Finally, given the smaller effect size for the nonword *cheechee* in Experiments 1a and 1b, we ran versions of the present analysis excluding trials with this item. There was still no significant effect of nonword type, nor a significant interaction between nonword type and age (whether treated as a categorical or a continuous variable; $p$'s > .23). See S7 and S8 Tables in S1 File.

We proceeded with planned comparisons at each visit separately. These analyses only included nonword type as a predictor. Again, no significant effect was found at visits one ($p$ = .87), two ($p$ = .93) and three ($p$ = .09). Bayesian analyses revealed that 73.45%, 71.85%, and 24.27% of the HDIs for nonword type, respectively, fell within the ROPE.

**Individual differences.** We next performed exploratory analyses to investigate if individual difference measures were related to the emergence of sensitivity to sound symbolism.

**Table 3. Linear mixed effects regression model predicting looking time for all infants with nonword type and age as a continuous variable.**

| Fixed Effect | b | SE | t | p |
|---|---|---|---|---|
| (Intercept) | -0.07 | 0.01 | 12.52 | $< .001$*** |
| Nonword Type | 0.01 | 0.01 | 0.69 | .49 |
| Age | 0.01 | 0.01 | 2.15 | 0.04* |
| Nonword Type x Age | 0.01 | 0.01 | 1.03 | 0.30 |
| Random Effect | $s^2$ | | | |
| Participant Intercept | 0.0003 | | | |
| Participant Age Slope | 0.0005 | | | |

Because there was solely a trend towards sensitivity to sound symbolism at visit three, we only analyzed individual difference measures from that visit. For each Developmental Questionnaire item, the answers were dichotomozied: 0 = child does not have a given ability; 1 = child has the given ability. A full description of this coding, as well as survey data from visits one and two, can be found at: https://osf.io/djgfx. We only analysed a subset of variables from the full survey, choosing those that we judged to best operationalize potential developmental milestones: experience producing speech sounds, language knowledge and environmental experience. To operationalize experience producing speech sounds, we performed an exploratory factor analysis on babbling data at visit three. Because these data were binary, we performed factor analysis using tetrachoric correlations. We began with babbling scores for the eight phonemes used in our study (i.e., /b, l, k, tʃ, oʊ, u, i, eɪ/). However, we excluded /b/ because all babies were babbling this speech sound, as well as the phonemes /u/ and /i/ because most babies were babbling these speech sounds (> 90%). A parallel analysis suggested reducing these variables to a single factor, see S9 Table in S1 File. We operationalized language knowledge as the understanding and productive scores on the MCDI (entered separately). Finally, we operationalized environmental experience by performing an exploratory factor analysis on the following physical milestones: walking with support, walking alone, and going up and down stairs. Parallel analysis once again suggested a single factor, see S10 Table in S1 File. This created five individual difference measures: a babbling factor, a vocal play score, an MCDI understanding score, an MCDI production score, and an environmental experience component.

Next, we operationalized sensitivity to sound symbolism at visit three as the average percent looking time to the congruent shape across both round and spiky trials. We also calculated the change in this variable from visit two to visit three, to quantify an increase in the sensitivity to sound symbolism. See Table 4 for a summary of all variables. We then computed eight correlations, one for each of these outcome variables and our individual differences, see Table 5. Only the correlation between the babbling factor and sensitivity to sound symbolism at visit three was significant ($r = .31$, $p = .045$), such that those who babbled more of the speech sounds from our nonword stimuli showed a larger effect. See Fig 5. However, this relationship would

**Table 4. Summary statistics of individual difference measures at visit three.**

| Binary Milestone | Percentage Achieved |
|---|---|
| Percentage Babbling /b/ | 100% |
| Percentage Babbling /l/ | 53.66% |
| Percentage Babbling /k/ | 52.8% |
| Percentage Babbling /tʃ/ | 39.2% |
| Percentage Babbling /oʊ/ | 80.95% |
| Percentage Babbling /u/ | 92.86% |
| Percentage Babbling /i/ | 90.8% |
| Percentage Babbling /eɪ/ | 85.1% |
| Percentage Walking with Support | 76.19% |
| Percentage Walking Alone Easily | 24.39% |
| Percentage Traversing Stairs | 61.90% |
| Continuous Meausre | Mean (SD) |
| MCDI Understanding out of 89 | 16.73 (10.57) |
| MCDI Production out of 89 | 1.34 (1.74) |
| Sound Symbolism Effect (Look Percentage to Congruent Minus Incongruent) | 1.47 (4.09) |
| Change in Effect Visit 2 to Visit 3 | 1.18 (5.72) |

**Table 5. Correlations between individual difference measures at visit three and sound symbolism measures.**

| | Sound Symbolism Sensitivity at Visit Three r (p) | Change from Visit Two to Visit Three r (p) |
|---|---|---|
| Babbling Factor | .31 (.045) | .28 (.08) |
| Vocal Play | .02 (.90) | -.00 (.99) |
| MCDI Understanding | -.14 (.37) | -.03 (.84) |
| MCDI Production | -.01 (.96) | -.14 (.39) |
| Movement Factor | .19 (.23) | -.00 (.98) |

not survive correction for multiple comparisons. The babbling factor was not correlated with the increase in sound symbolism from visits two and three ($r = .28$, $p = .08$).

## Discussion

We did not find a maluma/takete effect in our sample as a whole; nor at 4, 8 or 12 months of age. There was a trend towards this effect emerging at 12 months, however it was not significant. We found a moderate correlation between babbling experience and sensitivity to sound symbolism at 12 months, but this would not survive correction for multiple comparisons.

## Experiment 3

### Methods

**Participants.** We recruited a separate group of 71 infants ages 7–8 months ($M = 7.89$; $SD = 0.30$) again from the Ch.I.L.D. participant database at the University of Calgary and with

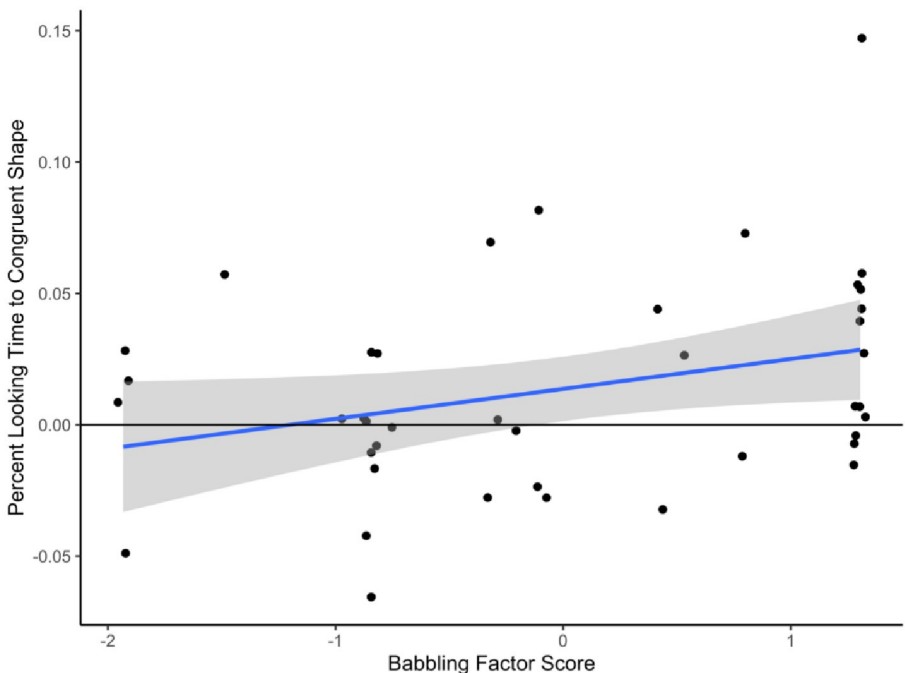

**Fig 5. Correlation between babbling factor score and percent looking time to congruent shape at visit 3.** $N = 43$.

a minimum 50% exposure to English; 26 were exposed to another language (Mandarin, Russian, Cantonese, Ukrainian, Japanese, French, Spanish, Farsi, Yoruba, Polish, Czech, German, Punjabi). Infants' parents or guardians provided written informed consent. Of these 71 infants, 11 were excluded from the analysis due to technology error, fussiness, dropping out of the study, insufficient English exposure, being too old for the study by the time they were tested, or incorrect label pronunciation (by caregiver when prompted during the second visit). In line with the target sample size, there were 30 infants per condition, with 15 per congruent and incongruent labels. This target was based on comparable studies in the literature (see Experiment 2) as well as practical limitations given the demanding nature of the study.

**Materials.**   In the preferential looking task we used the stimuli from Experiment 2 and we also included two new items. This consisted of a new round nonword (/mumu/) and spiky nonword (/titi/). Each of these was presented twice in the task during the second visit.

**Procedure.**   The procedure was similar to that described for Experiment 2, except that here families came for two visits about two weeks apart, and completed a learning task in between visits. During the first visit, caregivers completed the demographic and language measures and the infant completed the preferential looking task used in Experiment 2. At the end of the first visit, families were given a learning task to complete with their infant. That is, they were given a 3D object (Fig 6; either round or angular, and in one of the two colours, with a string attached) to take home. Families were instructed to expose their infant to the object for 10 minutes a day, saying the label a minimum of 12 times, for 14 days. A member of the research team pronounced the nonword to caregivers, and had them repeat it. Caregivers were instructed to not leave the infant alone with the object to ensure that the child's experience with the object always involved exposure to the label.

Families returned to the lab approximately two weeks after their first visit, give or take one or two days. During this second visit, a research assistant asked caregivers about how their learning task went, checking whether the instructions were followed and asking about the way they interacted with their infant during the learning task. In addition, caregivers' nonword pronunciation was checked, and their data were excluded if their pronunciation was incorrect. Next, families were once again seated in our testing room. Testing conditions were identical to the first visit except that the second visit included four extra trials with novel nonwords (i.e., [mumu] and [titi]). These additional trials allowed us to test whether infants learned the sound-symbolism bias and could apply it to novel labels as well as familiar ones.

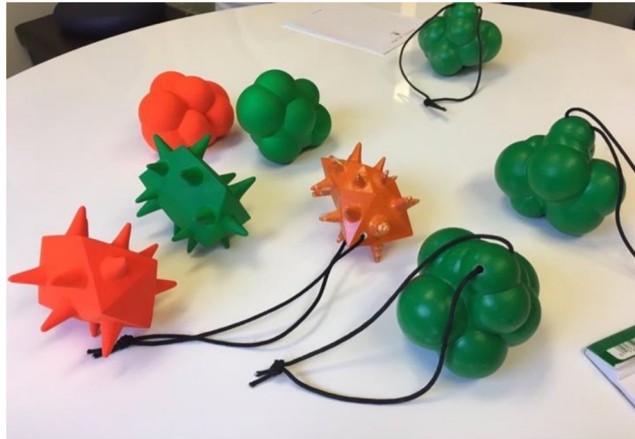

**Fig 6. 3D printed objects that families took home.** All finished objects had strings attached.

**Table 6. Linear mixed effects regression model predicting looking time for all infants with visit, training congruence, and their interaction.**

| Fixed Effect | b | SE | t | p |
|---|---|---|---|---|
| Intercept | -0.00 | 0.03 | -0.003 | .99 |
| Visit | -0.00 | 0.00 | -1.66 | .10 |
| Training Congruence | 0.02 | 0.01 | 1.45 | .15 |
| Visit x Training Congruence | -0.00 | 0.01 | -0.26 | .80 |
| Random Effect | $s^2$ | | | |
| Item Intercept | 0.01 | | | |

## Results

Analyses consisted of linear mixed effects regressions. Our dependent variable was percent of looking time to the congruent object. We centered this variable by subtracting 0.5 in order to allow interpretation of the intercept. Our initial analysis included as predictors: visit (after training [+.5] vs. before training [-.5]), training congruence (congruent [.5] vs. incongruent [-.5]), and their interaction. Table 6 summarizes the results. The intercept was not significantly different from 0 ($b = 0.00$, $p = .99$), indicating that participants did not show a preference for the congruent shape overall. We found no significant effects of visit ($b = -0.01$, $p = .10$) and training congruence ($b = 0.02$, $p = .15$). The interaction between predictors was not significant ($b = 0.00$, $p = .80$). See Fig 7. A Bayesian analysis revealed that 91.45% of the HDI for the interaction between visit and training congruence fell in the ROPE. As in Experiment 2, we ran a

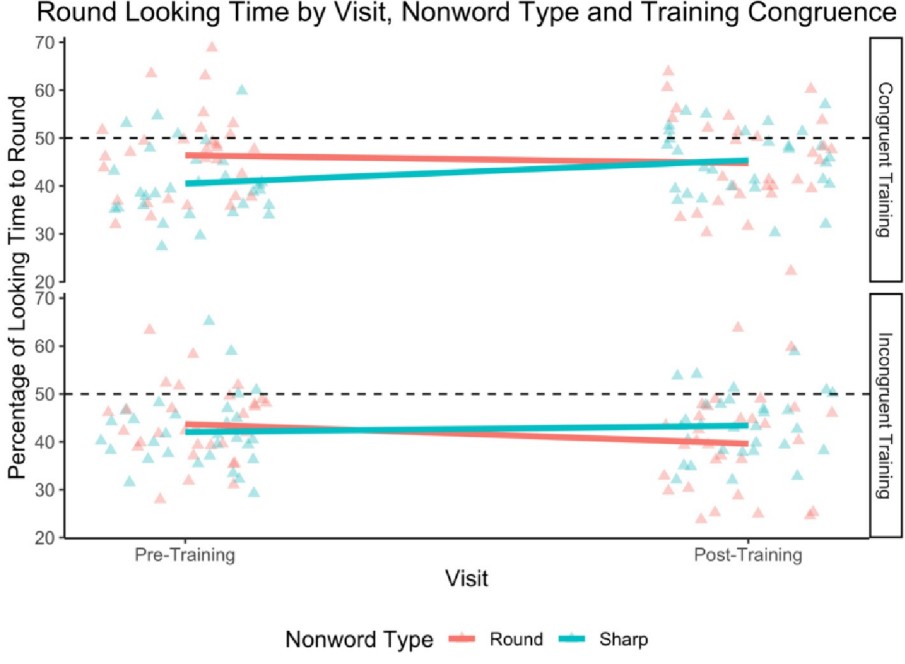

**Fig 7. Percent round object looking time by visit, nonword type, and training congruence, in Experiment 3.** This plot shows the percentage of total looking time spent looking at the round object, based on the visit, the type of nonword presented on each trial, and whether the participant was in the congruent or incongruent training condition. The dashed line corresponds to equal looking time between round and spiky shapes. $N = 60$.

version of this analysis excluding trials with the nonword *cheechee*. There was still no significant effect of visit nor a significant interaction between visit and congruence ($p$'s > .12).

We conducted a supplementary analysis to examine whether participants showed any evidence of learning the trained nonword. This analysis used the trials on which participants heard the nonword that they had received in training, and used the percent of looking time to the trained shape as the dependent variable. We used visit as the predictor of interest. Visit was not a significant predictor ($b$ = -0.00, $p$ = .74). A Bayesian analysis revealed that 86.25% of the HDI for Visit fell in the ROPE. This suggests that participants did not learn the nonword-shape pairing. We also examined if participants looked longer at the screen on trials where the colour of the shapes matched the trained shape and found that they did not ($b$ = -13.04, $p$ = .64). A Bayesian analysis revealed that 85.5% of the HDI for the familiar colour predictor fell in the ROPE. Note that in both of these instances we were able to include a random intercept for family in the Bayesian analysis, but not the frequentist model.

## Discussion

We did not find evidence that experience with a sound symbolically congruent label affected sensitivity to sound symbolism. Nevertheless, we should be cautious when interpreting the results from this study, as there was no evidence of training having an effect (no indication of greater attention to the take home object when hearing the trained nonword).

## General discussion

After verifying that the maluma/takete effect could be observed in adults using our novel shape stimuli (Experiments 1a and 1b), we conducted the first longitudinal study of infant sound symbolism (Experiment 2). This consisted of a looking time preference task comparing attention to a round and spiky shape while infants heard either a round- or a spiky-sounding nonword. Infants were tested at 4, 8 and 12 months. We found no evidence of sensitivity to sound symbolism: no difference in infants' attention to round and spiky shapes based on auditory stimuli. This was true overall, and also when the data from each visit were analyzed separately. There was, however, a trend towards sensitivity at 12 months. In addition, a factor representing individual babbling ability was moderately correlated with this effect. A subsequent study (Experiment 3) in which parents presented 8-month-old infants with congruent or incongruent labels for 3-D printed shapes also found no evidence of sensitivity to sound symbolism following training, for either label condition.

The first conclusion of note is that sensitivity to sound symbolism does not appear to be innate, nor present in the first few months of life. This is consistent with previous work which found no evidence of sound symbolism in four month old infants [31]. Some researchers have suggested that the neonatal brain contains widespread connectivity among sensory regions, which then become pruned during development (reviewed in [35]). In this view, some connections are believed to persist in typical adults, resulting in crossmodal associations such as the maluma/takete effect. Spector and Maurer [57] suggest that ". . .cross-modal effects similar to those seen in adult synesthesia are expected to occur during early childhood and to persist in muted form even in typical adults" (p. 177). This theory would motivate a prediction that sound symbolism will be present in very young infants. The present results do not support this. This is in contrast with other, non-linguistic, crossmodal associations have been observed at a young age: between visual elevation and sharpness at 3–4 months [58], between pitch and size at 6 months (but not 4 months; [59]), and between loudness and brightness at 20–30 days [60]. Perhaps the involvement of the linguistic system in sound symbolism leads to a slightly different etiology than that of other crossmodal associations (see [5]; cf. [61]).

Although we only observed marginal sensitivity to sound symbolism at 12 months, other studies have found an effect in infants of that age [14, 31, 32]. If this is indeed the age by which sound symbolism emerges, what factors might contribute to its development? In the Introduction we proposed several possibilities. One of these was language experience. Fort et al. [35] discussed the possibility of language acting as a "revelator" of sound symbolism. That is, infants may be born with the capacity for sound symbolic sensitivity, but experience with sound symbolic words in language serves to "reveal" (or perhaps highlight) such associations for an infant. This may explain differences in sound symbolism observed in adult speakers of certain languages [8]. However, note that exposure to sound symbolic language in Experiment 3 (i.e., those in the congruent label condition) did not increase this sensitivity. In addition, we did not find any relationships between infants' receptive or productive vocabulary and their sensitivity to sound symbolism. There was a floor effect in our measure of productive vocabulary ($M = 1.34$), and so it is possible that vocabulary could play a role later in development when infants' vocabulary is more substantial.

Another possibility was that environmental experience may play a key role in the development of sound symbolism. This relies on the assumption that round and spiky objects in the world tend to make different sounds, and that these sounds mimic round- and spiky-associated speech sounds [6, 26]. There is some tangential evidence for the role of environmental experience in sound symbolism. Work with visually- as well as hearing-impaired individuals suggests a sensitive period in which perceptual experience is key to the later development of sound symbolism [62, 63]. One might posit that this perceptual experience comprises patterns in the environment. Yet we did not find any evidence that the ability to explore the environment was related to sound symbolic sensitivity. It is important to note, however, that this is a rather coarse measure of environmental experience. Even before they begin to move themselves, infants are moved through the world by caregivers, and have objects brought to them by caregivers. A more direct way to test this possibility could be to track an infant's daily experience in the environment (e.g., [64]).

We did find that babbling ability was related to sound symbolism. Infants who babbled a greater number of speech sounds tended to show a greater sensitivity to sound symbolism. This could suggest that articulatory experience is a key contributor to the effect. Indeed, the role of oral motor movements in early infant speech perception is only beginning to be understood (e.g., [65]). Choi et al. recently demonstrated that oral motor interference diminished speech discrimination in 3-month-old prebabbling infants. Infants who have more babbling experience may also have more oral motor experience and thus more sensitivity to the sound-shape associations involved in the maluma/takete effect. This could be tangentially related to the cross-cultural finding that the maluma/takete effect may not emerge when the speech sounds in a nonword are not present in a speaker's language, or are in phonotactically illegal positions [8]. Although the correlation we observed would not survive correction for multiple comparisons, it does suggest an intriguing topic for future research. This should involve a longitudinal study with a larger sample size than the one reported here, in order to capture more individual variation. Such a study could also include more sensitive neuropsychological measures of sound symbolism (as in [14]).

What other developmental milestones, not captured here, might explain the emergence of sound symbolism? It may be that some specific neurological development is necessary. Indeed, maturation of the association cortex (including areas implicated in sound symbolism; [32, 34]) occurs later than individual sensory areas [66]. Work from individuals with diagnosis of Autism Spectrum Disorder and dyslexia has suggested that a key ability in sound symbolism is multisensory integration [67–69]—this could be another developmental milestone. Future work might also measure specific language milestones, such as the development of phonological

awareness, the ability to process metaphor or referential insight. As a way of attempting to capture this latter construct, we ran a supplementary analysis to see if a participant's looking time preference to the shape they were trained on while hearing their trained label at visit 2 of Experiment 3, predicted their sensitivity to sound symbolism, it did not ($b$ = -0.01, $p$ = .39).

It is worth exploring why we did not observe a robust sound symbolism effect at 12 months when other studies have done so [14, 31, 32]. One relevant difference may be that each of these previous studies used a single image task, while we presented infants with pairs of images. Note that Fort et al. [70] also found no evidence of the maluma/takete effect in 5- and 6-month-olds, using a looking time preference task with pairs of images. This led them to speculate that very early sound symbolism may only be detectable with simpler designs (i.e., with a single presented image, re-using the same round and spiky images on each trial). In addition, we only observed a modest sound symbolic effect for our spiky nonwords/images with adult participants in Experiments 1a and 1b. Using pictures of real objects may have made the curves and spikes in our stimuli less salient than in the simple and more abstract figures typically used [cf. 43]. Compare the outlines of the shapes in Fig 1 with those in Fig 2, for example. It is also worth considering that looking time may not be the ideal measure of sensitivity to sound symbolism. However, we are unaware of alternate measures that could be used across the range of ages tested here. At minimum, we can conclude that infant sound symbolism is not an especially robust phenomenon, and that it is sensitive to methodological details, the specific influence of which is not yet well understood.

## Conclusion

We did not find evidence of sound symbolism in infants at 4, 8 or 12 months of age. However, consistent with other work Pejovic and Molnar [31], there was a trend towards sound symbolism emerging at 12 months. This emerging sensitivity correlated with the development of babbling in individual infants, but not with their ability to explore the environment or their vocabulary. Further investigations of individual differences in the development of sound symbolism could be a fruitful area for future research.

## Supporting information

**S1 File.**
(PDF)

## Acknowledgments

The authors would like to thank Dr. Ford Burles for creating the 3D objects used in these studies, Christopher Engelking for helping with data management, and Summer Abdalla and Julia Kim for helping with data collection.

## Author Contributions

**Conceptualization:** David M. Sidhu, Angeliki Athanasopoulou, Stephanie L. Archer, Natalia Czarnecki, Suzanne Curtin, Penny M. Pexman.

**Data curation:** David M. Sidhu, Angeliki Athanasopoulou, Natalia Czarnecki.

**Formal analysis:** David M. Sidhu.

**Funding acquisition:** David M. Sidhu, Stephanie L. Archer, Suzanne Curtin, Penny M. Pexman.

**Investigation:** David M. Sidhu, Angeliki Athanasopoulou, Stephanie L. Archer, Natalia Czarnecki, Suzanne Curtin, Penny M. Pexman.

**Methodology:** David M. Sidhu, Angeliki Athanasopoulou, Stephanie L. Archer, Natalia Czarnecki, Suzanne Curtin, Penny M. Pexman.

**Project administration:** Angeliki Athanasopoulou, Stephanie L. Archer, Natalia Czarnecki, Suzanne Curtin, Penny M. Pexman.

**Resources:** Suzanne Curtin, Penny M. Pexman.

**Software:** Suzanne Curtin, Penny M. Pexman.

**Supervision:** Angeliki Athanasopoulou, Stephanie L. Archer, Suzanne Curtin, Penny M. Pexman.

**Visualization:** David M. Sidhu.

**Writing – original draft:** David M. Sidhu, Angeliki Athanasopoulou, Penny M. Pexman.

**Writing – review & editing:** David M. Sidhu, Angeliki Athanasopoulou, Stephanie L. Archer, Natalia Czarnecki, Suzanne Curtin, Penny M. Pexman.

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
