## [Decision Letter · Decision Letter 0]

7 Mar 2023

PONE-D-22-33843The Maluma/Takete Effect is Late:

No Longitudinal Evidence for Shape Sound Symbolism in the First YearPLOS ONE

Dear Dr. Sidhu,

Thank you for submitting your manuscript to PLOS ONE. After careful consideration, we feel that it has merit but does not fully meet PLOS ONE’s publication criteria as it currently stands. Therefore, we invite you to submit a revised version of the manuscript that addresses the points raised during the review process.

I am thankful to have now received reviews from two experts in bouba/kiki sound symbolism. Both reviewers are highly positive about the paper, with Reviewer 1 recommending it be accepted and Reviewer 2 recommending minor revisions. Both reviewers offer a number of constructive suggestions for improving the paper, and Reviewer 2 in particular provided very detailed suggestions for making the data and analysis open and replicable. I encourage you to take both reviews into consideration as much as possible in your revision.

We look forward to receiving your revised manuscript.

Kind regards,

Marcus Perlman, Ph.D

Academic Editor

PLOS ONE

Journal Requirements:

2. Please provide additional details regarding ethical approval in the body of your manuscript. In the Methods section, please ensure that you have specified the name of the IRB/ethics committee that approved your study.

3. Please provide additional details regarding participant consent. In the Methods section, please ensure that you have specified (1) whether consent was informed and (2) what type you obtained (for instance, written or verbal). If your study included minors, state whether you obtained consent from parents or guardians. If the need for consent was waived by the ethics committee, please include this information.

"This research was supported by the Social Sciences and Humanities Research Council of Canada (SSHRC) through an Insight Development Grant (430-2017-00003). "

"This research was supported by the Social Sciences and Humanities Research Council of Canada (SSHRC; https://www.sshrc-crsh.gc.c) through an Insight Development Grant (430-2017-00003) to PP and SC. The funders had no role in study design, data collection and analysis, decision to publish, or preparation of the manuscript. "

6. PLOS requires an ORCID iD for the corresponding author in Editorial Manager on papers submitted after December 6th, 2016. Please ensure that you have an ORCID iD and that it is validated in Editorial Manager. To do this, go to ‘Update my Information’ (in the upper left-hand corner of the main menu), and click on the Fetch/Validate link next to the ORCID field. This will take you to the ORCID site and allow you to create a new iD or authenticate a pre-existing iD in Editorial Manager. Please see the following video for instructions on linking an ORCID iD to your Editorial Manager account: https://www.youtube.com/watch?v=_xcclfuvtxQ.

Reviewers' comments:

Reviewer's Responses to Questions

**Comments to the Author**

1. Is the manuscript technically sound, and do the data support the conclusions?

Reviewer #1: Yes

Reviewer #2: Yes

2. Has the statistical analysis been performed appropriately and rigorously? 

Reviewer #1: Yes

Reviewer #2: Yes

3. Have the authors made all data underlying the findings in their manuscript fully available?

Reviewer #1: Yes

Reviewer #2: Yes

4. Is the manuscript presented in an intelligible fashion and written in standard English?

Reviewer #1: Yes

Reviewer #2: Yes

5. Review Comments to the Author

Reviewer #1: This paper was well written and makes a valuable contribution to our understanding of the development of sound symbolism and it's potential relationship to language acquisition. The methods and results especially are very rigorous and clearly explained. I look forward to seeing this in print. I have some minor suggestions the authors may want to consider:

p.3, second paragraph: This makes it seem as if analogy is the *only* possible mechanism for sound symbolism; I'd make it clearer that this is one example of a mechanism that's relevant to the work in this paper (it's distinct e.g., from the frequency code hypothesis). No need to get into it in detail, maybe refer the reader to existing ref 21 here

p.3 Footnote 1: It's worth referring the reader to a more detailed treatment here; maybe Ortega (2017, https://doi.org/10.3389/fpsyg.2017.01280)? But there may be better options...

p.5, lines 88-98: It's worth noting somewhere in this paragraph that none of these theories are necessarily mutually exclusive - it seems the question is the degree to which each of these things may play a role (and this may change over the course of development).

p. 20, lines 387-394: "Because participants could only look at the round or the spiky object..." I'm not that familiar with coding in preferential looking paradigms, so I might be missing something, but isn't it possible that they spent some time looking at neither? Was that time (which was presumably <2s anyway) just excluded from this calculation? Or were looking times for a stimulus <2s excluded? This isn't quite clear.

p. 30, lines 524-529: "26 were exposed to another language...Of these infants, 11 were excluded..." This phrasing sounds like 11 of the 26 were excluded, when I expect it's 11 of the original 71 (some of whom may have met the "exposed to another language" criteria). It would make more sense to report this exclusion first, then report the language exposure, and maybe close out the paragraph by confirming that there were 30 in each condition ("The target sample size was 30" seems aspirational, but this is what you presumably did, e.g. "In line with the target sample size, there were 30 infants per condition")

p. 35, lines 604-608: "The first conclusion of note is that sensitivity to sound symbolism does not appear to be innate." While this is certainly true, I'm not sure this is a serious hypothesis anyone really has? What would it mean for it to be innate, exactly? (see e.g., Mameli & Bateson, 2011). I see the point re: the neonatal synesthesia hypothesis directly after this, but I'm not sure this lands either; if there's widespread connectivity during development, couldn't some form of this theory *predict* that young infants wouldn't yet prefer the *specific* mappings adults show preferences for (because this comes with pruning). In other words, I'm not sure this theory does predict "sensitivity to cross-modal associations like maluma takete"; it predicts a general capacity for any cross-modal mapping at a particular stage in development.

p. 35, lines 620-622: "Infants may be born with sensitivity to sound symbolic relationships, but experience with sound symbolic words in language serves to "reveal" specific associations". This claim seems compatible with the neonatal synaesthesia hypothesis, and so seems to contradict the previous paragraph. Some light rewrites on this page to bring clarity on how you're interpreting this theory (and how this claim relates to lines 604+) would go a long way.

Discussion around vocabulary: Can you address the possibility that your vocabulary measures might be too low across the board to catch any effect, e.g., perhaps vocabulary level does predict sound symbolic associations, but this effect is later?

Discussion around referential insight: Can you unpack what this is and point to some other work on this? When does this capacity generally develop? Might be worth unpacking footnote 12?

General Discussion: Is it possible that preferential looking just...isn't a very reliable measure of sound symbolic associations? This seems a good potential explanation for the mixed results in the literature. With infants this young options are obviously limited, but it seems prudent to discuss the possibility that the measure here is not capturing what we want.

Signed: Christine Cuskley

Mameli, M., & Bateson, P. (2011). An evaluation of the concept of innateness. *Philosophical Transactions of the Royal Society B: Biological Sciences*, *366*(1563), 436-443.

Reviewer #2: This article describes a series of studies bouba/kiki eye-gaze studies designed to document developmental changes in children between the ages of 4 and 12 months. The studies are well designed and adequately powered to detect both main effects of sound symbolism, and changes in the strength or direction of such an effect over time. The stimuli were validated as suitable bouba/kiki type sound symbolic matching in a sample of adults. A separate test was conducted to establish gaze properties of the stimuli in the age range of interest. In two critical hypothesis tests, children did not show any bouba/kiki effect (i.e., different gaze for congruent versus incongruent pairings of visual and audio stimuli), and this pattern did not differ with age, nor with additional exposure to objects and their labels. These findings differ from previously published reports of gaze behavior consistent with bouba/kiki effects within this age range, making this report an important contribution to the field. The authors discuss possible reasons for the lack of clear bouba/kiki responding, including highlighting the possibility that sound symbolic matching may take some time to emerge alongside developing language skills.

Given the extremely small number of published bouba/kiki effects in developmental samples, this report is important, timely, and adds considerably to our knowledge of how, whether and when young children might show evidence of sound-symbolic congruence. Indeed, many researchers in this field have heard myriad whispers of ‘failed’ bouba/kiki eye-gaze tests that have never been reported, suggesting a substantial file drawer of unpublished results. Hence, despite the data aligning with the null hypothesis (i.e., no evidence of effect), the lack of evidence is informative – particularly as the study includes a series of tests. The report is clearly written, and the data are archived in an open access repository. I recommend publication with minor corrections only to enhance the clarity and completeness of the report and repository.

Theory/Lit review/Discussion

Page: 10

The lit review suggests the bouba/kiki effect occurs for 90% of participants in all language contexts, even though one of the cited papers in this section (Styles & Gawne 2017) proposes that the 'canonical' word forms do not work in all language contexts. This means that it may be shaped by language exposure across development. The authors do return to this point in the discussion, but there might be a way to foreshadow this point in the introduction. Consistent with this view: work by (Shang & Styles, 2017, 2023) shows that the effect can take a different form in different languages, and Woon (2018) showed that children in early primary school show a weaker bouba/kiki effect than adults in a large developmental sample.

Styles, S. J., & Gawne, L. (2017). When does maluma/takete fail? Two key failures and a meta-analysis suggest that phonology and phonotactics matter. i-Perception, 8(4), 1-17. doi:10.1177/2041669517724807

Shang, N., & Styles, S. J. (2017). Is a High Tone Pointy? Speakers of Different Languages Match Mandarin Chinese Tones to Visual Shapes Differently. Frontiers in Psychology, 8, 2139. https://doi.org/10.3389/fpsyg.2017.02139

Shang, N., & Styles, S. J. (2023). Implicit Association Test (IAT) Studies Investigating Pitch‐Shape Audiovisual Cross‐modal Associations Across Language Groups. Cognitive Science, 47(1). https://doi.org/10.1111/cogs.13221

Woon, F. T. (2018). Linguistic sound symbolism and reading development: Sound-shape matching and predicters of reading in multilingual Singapore. (Masters), Nanyang Technological University. https://hdl.handle.net/10356/73469

Page: 11

I think Ramachandran was the first to make the proposal about congruence of the movement of articulators in the bouba/kiki effect (although phrased rather vaguely). He was certainly very clear about the round shape of the lips though, so a reference to lip shape should either include his contribution, or a clear statement that more recent papers are summaries/overviews.

Ramachandran, V. S., & Hubbard, E. M. (2001). Synaesthesia—A Window Into Perception, Thought and Language. Journal of Consciousness Studies, 8(123-34).

Ohala was the first to describe the relationships between the shapes of objects and the sounds they make in the natural world. These relationships are noted in the discussion without reference to Ohala.

Ohala, J. J. (1994). The frequency code underlies the sound-symbolic use of voice pitch. In L. Hinton, J. Nichols, & J. J. Ohala (Eds.), Sound Symbolism (pp. 325-347). Cambridge: Cambridge University Press.

In general, the discussion of the paper could make clearer the different possibilities for why their data do not align with the previously published bouba/kiki effects in children under 12m. If the current study is a false negative, the bouba/kiki effect may be true of the general population, but the failure to detect the effect here may be simply unlucky, OR it may reflect differences in implementation of the task (differences in stimuli, differences in test paradigm, differences in some characteristic of the population). However, the other possibility is that the current study reports a true negative, and previous results may represent false positives – there might be a robust, replicable bouba/kiki effect that is generally true of the population of children under 1 year. If this is the case we would expect there to be other (perhaps unpublished) studies which also failed to observe a significant effect in early developmental samples. This possibility is hinted at in the discussion, as developmental mechanisms are suggested which might account for the gradual emergence of the effect, but the point could be made even more clearly. Styles & Gawne (2017), for example, highlight a possible ‘file drawer’ of failed studies, and encourage other authors to “bring out their dead” (p.12). If the authors believe that other explanations are more likely (e.g., individual differences in babbling across samples), the discussion could propose what particular programme of research would provide confirmatory or discriminatory evidence. This kind of clarification would help to strengthen the theoretical contribution of the paper, and provide a concrete pathway to future research on the topic.

At the moment the discussion and closing statements of the article focus instead on the (somewhat underpowered) correlation between individual babbling and possible emergence of sound symbolic looking behaviour. While this is a tantalizing possibility, it would require future research to clarify whether this pattern is replicable in a well powered sample. The authors could be a bit more specific about precisely what kind of research would be needed to confirm this kind of individual difference.

Methods

Not sure what transliteration is being used by the team to describe the pronunciations. In the International Phonetic Alphabet the /o/ in /bobo/ is the monopthong vowel in 'got' NOT the dipthong vowel in 'go'. I think this might be the wrong symbol if the 'written label is 'boeboe'. If the word rhymes with 'logo' and 'oboe' then the symbol should be something like /ou/ or /oʊ/. Similar problem for the other vowels I think, but hard to be sure. This is a really common problem in bouba/kiki type research, and it makes replication and metaanalysis really hard. Please check your symbols align with the International Phonetic Alphabet, and/or to describe the stimulus words using unambiguous rhymes (e.g., 'oboe'). Where can a reader hear the audio tokens used in the study? I couldn't find the audio recordings in the OSF repository to check.

Did the parents ever hear a member of the research team say the words aloud before the toys were sent home, or did they first encounter them written down? One brief statement suggests a parent's pronunciation was checked to for alignment to the expected wordform, but this isn’t described clearly – the procedure for the check could be added to the OSF.

Reporting of results

• Percentages reported without error-variance/confidence interval. Please add an index of confidence around point estimates or provide the raw frequency data in the table so the confidence can be estimated without referring to the data in supplements.

• Please include effects sizes for all effects and interactions – even when non-significant, as this is more consistent with full reporting. In some places, non-significant p-values are summarized (e.g., all p>.33). I believe most of these are reported in full in tables, but please check, as I could not find full reporting of supplementary analyses.

•Please report supplementary analyses in full. Results of key models are tabulated in text, along with mention of supplementary analyses in the text/ footnotes. If I understood correctly, the additional analyses are not reported/tabulated in full anywhere. The full analysis script is provided along with the data for analysis, but this requires a reader to run the analysis script to view the full statistical report. As the current software may become obsolete in the future, please add a supplementary analysis document to the OSF repository where the full statistical outputs can be viewed.

Statistical Models

Please clarify - is the 'simple' regression after removing random participant intercept still a repeated measures design? This is important because one person's responses may be correlated more closely to their own responses than to others' responses - even without the random effect being significant (i.e., people respond moderately differently to one another, but not significantly). By keeping the random factor in your model, you may actually improve the power to detect an effect. Perhaps there is different reasoning behind this decision, so I'd be happy to understand more about it - may be in a footnote or supplementary materials.

Graphing

Best practice in data visualization recommends that every figure summarizing the results of more than one participant include the participant N on the plot (e.g., in the legend) or in the

Figure description, so that a reader doesn't have to go searching for the info in other parts of the article. Please add N to each graph or each age group. This also helps a reader to get an overview of whether the graph includes incomplete data across longitudinal samples (e.g., more samples at earlier than timepoints) or includes only those samples who were tested across all timepoint (i.e., no dropouts included in graph).

Best practice in data viz is to include individual data points where possible, as well as an indication of confidence estimates about any point estimate drawn from more than one data point. Figure 3 uses bars to represent percentage of responses. This obfuscates the precise computation from 8 trials down to four bars: are we seeing the average value across the group for the individuals’ percentages in each condition (i.e., only possible values for each person are 100%, 50%, 0%)? Or something else? The authors could consider an alternative way of graphing to show individual binary responses – for example, dot plots - this would help to visually communicate the structure of the data, and make clearer that participants responses differ substantially across different items (e.g., almost at chance for ‘cheechee’)

Archive of Data, Code and Analysis

Code for running the studies not yet provided in OSF. Please provide the scripts for the adult study (EPrime) and the eye-gaze studies (Habit). Including these would help make the study (or future adaptations) more replicable in the future.

3D wireframes. Since the test objects were 3d printed, it would be great to include the wireframes used for 3D printing of the objects, or a link to a digital file on a maker-site. This would enhance future replicability of the project.

Visual Stimuli For future replicability, please add photographs of the 3D shapes as used on screen in the study.

Audio Stimuli. One of the challenges of bouba/kiki studies internationally is knowing precisely what participants heard. One solution to this problem is accurate transcription in IPA, but as became clear in the metaanalysis by Styles & Gawne (2018), the notations provided in papers do not always align with strict IPA transcription. To allow greater transparency, please document the audio files in the OSF. As audio recording quality can differ substantially, including the audio files would readers to know what participants actually heard, including whether the content of the audio files can be easily heard by study participants. Archiving would also allow potential replicability into the future.

Data files in archive have several different structures. It would be super helpful to include a code-book or a wiki explaining the different data files, their origin (what software & version?), and what each column represents in each file. Some of the Habit files share a structure, but for a casual viewer, it is important to know which columns are user-defined (e.g., Family and ID), and which of the automated columns are used in analysis. The Individual differences data file is particularly rich, but it is not clear where to look for the coding scheme if you start by looking at the data (the survey form I think?).

Surveys

Current archive only includes the survey from T1, but the text mentions that later surveys have different questions about babbling. Please include the full set of surveys for the different timepoints, with the precise questions answered by parents, and any protocol for implementation if additional questions were asked by a researcher.

• The archive appears to only include data from T3 – which appears in the report – but as data were collected at T1 and T2, it would be better to include a complete record of survey data at all three timepoints. Alternatively, if those data are reserved for another project, this should be described along with a link to where the data are archived separately. OSF allows archives to be created before being made public so if an embargo currently exists, a link can be provided for future accessibility.

• Many of the questions in the survey are not reported in the current writeup. There can be many reasons for not reporting all outcome measures, including that the bouba/kiki study reported here was part of a larger research project, which will be reported in full elsewhere. It is important to disclose the purpose for collecting tis data so a reader can know whether there was more exploratory analysis than the results reported here (i.e., multiple possible outcome measures were collected and only significant ones are reported in the current paper) or pre-conceived hypothesis tests. If exploratory, the analysis should be reported as such, along with the results of all non-significant tests, so that the probability values can be understood in context. If the study was preregistered, please clarify this in the main text of the study description so that the prereg can be readily identified.

Minor points (see attached):

6. PLOS authors have the option to publish the peer review history of their article (what does this mean?). If published, this will include your full peer review and any attached files.

Reviewer #1: **Yes: **Christine Cuskley

Reviewer #2: **Yes: **Styles, Suzy J

---

## [Author Response · Author response to Decision Letter 0]

12 Jun 2023

Note. Response to reviewers also included in Cover Letter, with reviewer comments bolded to help with legibility.

Reviewer 1

p.3, second paragraph: This makes it seem as if analogy is the *only* possible mechanism for sound symbolism; I'd make it clearer that this is one example of a mechanism that's relevant to the work in this paper (it's distinct e.g., from the frequency code hypothesis). No need to get into it in detail, maybe refer the reader to existing ref 21 here

Thanks for this suggestion, we have noted that there are other potential mechanisms, and have added a reference to a review paper.

p.3 Footnote 1: It's worth referring the reader to a more detailed treatment here; maybe Ortega (2017, https://doi.org/10.3389/fpsyg.2017.01280)? But there may be better options...

Thank you, we have added this reference.

p.5, lines 88-98: It's worth noting somewhere in this paragraph that none of these theories are necessarily mutually exclusive - it seems the question is the degree to which each of these things may play a role (and this may change over the course of development).

Thank you we have added this.

p. 20, lines 387-394: "Because participants could only look at the round or the spiky object..." I'm not that familiar with coding in preferential looking paradigms, so I might be missing something, but isn't it possible that they spent some time looking at neither? Was that time (which was presumably <2s anyway) just excluded from this calculation? Or were looking times for a stimulus <2s excluded? This isn't quite clear.

We have now added a clarification that we removed “any time spent not looking at either shape from the analyses”. We also clarified that “We excluded trials on which a participant did not spend at least two seconds looking at either shape”.

p. 30, lines 524-529: "26 were exposed to another language...Of these infants, 11 were excluded..." This phrasing sounds like 11 of the 26 were excluded, when I expect it's 11 of the original 71 (some of whom may have met the "exposed to another language" criteria). It would make more sense to report this exclusion first, then report the language exposure, and maybe close out the paragraph by confirming that there were 30 in each condition ("The target sample size was 30" seems aspirational, but this is what you presumably did, e.g. "In line with the target sample size, there were 30 infants per condition")

We clarified this paragraph and now say: 

Of these 71 infants, 11 were excluded from the analysis…

In addition we have clarified the target sample size as: 

In line with the target sample size, there were 30 infants per condition, with 15 per congruent and incongruent labels. This target was based on comparable studies in the literature (see Experiment 2) as well as practical limitations given the demanding nature of the study.

p. 35, lines 604-608: "The first conclusion of note is that sensitivity to sound symbolism does not appear to be innate." While this is certainly true, I'm not sure this is a serious hypothesis anyone really has? What would it mean for it to be innate, exactly? (see e.g., Mameli & Bateson, 2011). I see the point re: the neonatal synesthesia hypothesis directly after this, but I'm not sure this lands either; if there's widespread connectivity during development, couldn't some form of this theory *predict* that young infants wouldn't yet prefer the *specific* mappings adults show preferences for (because this comes with pruning). In other words, I'm not sure this theory does predict "sensitivity to cross-modal associations like maluma takete"; it predicts a general capacity for any cross-modal mapping at a particular stage in development.

We agree that an innate effect seems extremely implausible, and that it is not well characterised in the literature. However, it is mentioned in the Symbouki review paper as a legitimate hypothesis (though not one that is upheld by the data). The authors say, regarding their meta-analysis, 

“This theory predicts that children show a strong sensitivity to cross-modal correspondences such as the bouba-kiki effect in their earliest stages of development, and that this sensitivity decreases over time through selective synaptic pruning. The present results are clearly not in accordance with these predictions. While the lack of an age effect for the bouba-type stimuli leaves open the possibility that the bouba effect is indeed innate, the overall modest effect size in the present data and the absence of a kiki effect in the earliest stages of development demonstrate that the overall bouba-kiki effect is weak before the age of 3. Moreover, the discrepancy observed be- tween bouba-type and kiki-type associations cannot be explained by the neonatal synesthesia theory as it is formulated by Spector and Maurer (2009). Indeed, there is so far no mechanism in this theory that can account for the pace difference in the emergence of a kiki and a bouba effect, respectively.” (Fort et al., 2018, p. 9). 

(Emphasis added.)

With this in mind, it does seem worthwhile to point out that our results argue against such an interpretation. However, we have rephrased slightly to “innate or present in the first few months of life”.

Our understanding is that the neonatal synesthesia hypothesis would still predict specific associations are present prior to pruning. Our understanding is that extra connections from, say, auditory to tactile regions would allow for round sound -> round shape associations. They wouldn’t mean that round sounds go with both round and spiky shapes until pruning. Spector and Maurer (2009) say “By either account, cross-modal effects similar to those seen in adult synesthesia are expected to occur during early childhood and to persist in muted form even in typical adults.” (p. 177). Thus, our understanding is that non-pruned connections provide the “infrastructure” for associations, but something else still determines which specific associations are present. We have rephrased slightly to make clearer how we are interpreting the relationship between pruning and adult CM associations. 

p. 35, lines 620-622: "Infants may be born with sensitivity to sound symbolic relationships, but experience with sound symbolic words in language serves to "reveal" specific associations". This claim seems compatible with the neonatal synaesthesia hypothesis, and so seems to contradict the previous paragraph. Some light rewrites on this page to bring clarity on how you're interpreting this theory (and how this claim relates to lines 604+) would go a long way.

Have tried to rephrase. Now suggesting that this theory proposes infants could be born with capacity for sensitivity to sound symbolism, but that sound symbolic language serves to reveal or highlight these associations for an infant. (This is in contrast to the previous wording which sounded like infants are born with all associations, and then language guides them to sound symbolic ones.)

Discussion around vocabulary: Can you address the possibility that your vocabulary measures might be too low across the board to catch any effect, e.g., perhaps vocabulary level does predict sound symbolic associations, but this effect is later?

Thank you for this point, we now note this in the General Discussion.

Discussion around referential insight: Can you unpack what this is and point to some other work on this? When does this capacity generally develop? Might be worth unpacking footnote 12?

We have changed the paragraph regarding referential insight and infant development. 

In their sound symbolism bootstrapping hypothesis, Imai and Kita [15] claim that sound symbolism provides infants with the insight that speech sounds refer to things in the world. For example,14-month-olds match novel objects with sound symbolic words (kipi, moma) after habituation to the sound symbolic pairings (Imai, Mizazaki, Yeung, Hidaka, Kantartzis, Okada, & Kita, 2014). However, if that is the case, then we should expect infants to demonstrate this type of insight much earlier.

General Discussion: Is it possible that preferential looking just...isn't a very reliable measure of sound symbolic associations? This seems a good potential explanation for the mixed results in the literature. With infants this young options are obviously limited, but it seems prudent to discuss the possibility that the measure here is not capturing what we want.

Your point is well taken. Unfortunately, as you say, the options are limited with young infants. Looking time preference tasks have been able to detect novel word/object associations (see our reply to a previous point), but it is entirely possible that it is not the ideal measure for sound symbolic associations. We now say as much in the final paragraph of the General Discussion:

It is also worth considering that looking time may not be the ideal measure of sensitivity to sound symbolism. However, we are unaware of alternate measures that could be used across the range of ages tested here.

Reviewer 2:

Page: 10

The lit review suggests the bouba/kiki effect occurs for 90% of participants in all language contexts, even though one of the cited papers in this section (Styles & Gawne 2017) proposes that the 'canonical' word forms do not work in all language contexts. This means that it may be shaped by language exposure across development. The authors do return to this point in the discussion, but there might be a way to foreshadow this point in the introduction. Consistent with this view: work by (Shang & Styles, 2017, 2023) shows that the effect can take a different form in different languages, and Woon (2018) showed that children in early primary school show a weaker bouba/kiki effect than adults in a large developmental sample.

Styles, S. J., & Gawne, L. (2017). When does maluma/takete fail? Two key failures and a meta-analysis suggest that phonology and phonotactics matter. i-Perception, 8(4), 1-17. doi:10.1177/2041669517724807

Shang, N., & Styles, S. J. (2017). Is a High Tone Pointy? Speakers of Different Languages Match Mandarin Chinese Tones to Visual Shapes Differently. Frontiers in Psychology, 8, 2139. https://doi.org/10.3389/fpsyg.2017.02139

Shang, N., & Styles, S. J. (2023). Implicit Association Test (IAT) Studies Investigating Pitch‐Shape Audiovisual Cross‐modal Associations Across Language Groups. Cognitive Science, 47(1). https://doi.org/10.1111/cogs.13221

Woon, F. T. (2018). Linguistic sound symbolism and reading development: Sound-shape matching and predicters of reading in multilingual Singapore. (Masters), Nanyang Technological University. https://hdl.handle.net/10356/73469

Thank you for this suggestion, we now note that there is variation based on language differences. We now also note that the idea of language as a revelator may explain differences observed in adult speakers of certain languages. 

Page: 11

I think Ramachandran was the first to make the proposal about congruence of the movement of articulators in the bouba/kiki effect (although phrased rather vaguely). He was certainly very clear about the round shape of the lips though, so a reference to lip shape should either include his contribution, or a clear statement that more recent papers are summaries/overviews.

Ramachandran, V. S., & Hubbard, E. M. (2001). Synaesthesia—A Window Into Perception, Thought and Language. Journal of Consciousness Studies, 8(123-34).

Thank you for pointing this out, we have added a reference to this paper. 

Ohala was the first to describe the relationships between the shapes of objects and the sounds they make in the natural world. These relationships are noted in the discussion without reference to Ohala.

Ohala, J. J. (1994). The frequency code underlies the sound-symbolic use of voice pitch. In L. Hinton, J. Nichols, & J. J. Ohala (Eds.), Sound Symbolism (pp. 325-347). Cambridge: Cambridge University Press.

Thank you for pointing this out, we have added a reference to this chapter.

In general, the discussion of the paper could make clearer the different possibilities for why their data do not align with the previously published bouba/kiki effects in children under 12m. If the current study is a false negative, the bouba/kiki effect may be true of the general population, but the failure to detect the effect here may be simply unlucky, OR it may reflect differences in implementation of the task (differences in stimuli, differences in test paradigm, differences in some characteristic of the population). However, the other possibility is that the current study reports a true negative, and previous results may represent false positives – there might be a robust, replicable bouba/kiki effect that is generally true of the population of children under 1 year. If this is the case we would expect there to be other (perhaps unpublished) studies which also failed to observe a significant effect in early developmental samples. This possibility is hinted at in the discussion, as developmental mechanisms are suggested which might account for the gradual emergence of the effect, but the point could be made even more clearly. Styles & Gawne (2017), for example, highlight a possible ‘file drawer’ of failed studies, and encourage other authors to “bring out their dead” (p.12). If the authors believe that other explanations are more likely (e.g., individual differences in babbling across samples), the discussion could propose what particular programme of research would provide confirmatory or discriminatory evidence. This kind of clarification would help to strengthen the theoretical contribution of the paper, and provide a concrete pathway to future research on the topic.

The authors are only aware of one study which has shown evidence of sensitivity to sound symbolism in infants younger than one year (Ozturk et al., 2008). Asano et al. (2015) did find evidence in infants who were 11 months using EEG, however their mean age was actually 11 months and 25 days. With this in mind, we would consider our results to be largely consistent with the literature testing infants less than one year.

To your latter point, we now suggest that a path forward investigating individual differences would ideally include a larger sample size, and potentially a more sensitive neuroimaging measure.

At the moment the discussion and closing statements of the article focus instead on the (somewhat underpowered) correlation between individual babbling and possible emergence of sound symbolic looking behaviour. While this is a tantalizing possibility, it would require future research to clarify whether this pattern is replicable in a well powered sample. The authors could be a bit more specific about precisely what kind of research would be needed to confirm this kind of individual difference.

See above. 

Not sure what transliteration is being used by the team to describe the pronunciations. In the International Phonetic Alphabet the /o/ in /bobo/ is the monopthong vowel in 'got' NOT the dipthong vowel in 'go'. I think this might be the wrong symbol if the 'written label is 'boeboe'. If the word rhymes with 'logo' and 'oboe' then the symbol should be something like /ou/ or /oʊ/. Similar problem for the other vowels I think, but hard to be sure. This is a really common problem in bouba/kiki type research, and it makes replication and metaanalysis really hard. Please check your symbols align with the International Phonetic Alphabet, and/or to describe the stimulus words using unambiguous rhymes (e.g., 'oboe'). Where can a reader hear the audio tokens used in the study? I couldn't find the audio recordings in the OSF repository to check.

Thank you for pointing this out, we have changed the transcriptions of two of our nonwords (i.e., to [ boʊboʊ] and [keɪkeɪ]), and in our report of the babbling questionnaire. We now also use square brackets instead of slashes, to indicate that these are transcriptions of the actual sounds heard rather than phoneme categories. 

We have also uploaded our audio stimuli to the OSF repository. 

Did the parents ever hear a member of the research team say the words aloud before the toys were sent home, or did they first encounter them written down? One brief statement suggests a parent's pronunciation was checked to for alignment to the expected wordform, but this isn’t described clearly – the procedure for the check could be added to the OSF.

Yes the research team said the words to the families, who repeated them back to the researchers. We also checked the parent/caregiver’s pronunciation when they returned to the lab. We have clarified this in the manuscript. 

• Percentages reported without error-variance/confidence interval. Please add an index of confidence around point estimates or provide the raw frequency data in the table so the confidence can be estimated without referring to the data in supplements.

Have added 95% CIs. However recall that subjects only saw each word twice, thus these values could only be 0, .5 or 1. 

• Please include effects sizes for all effects and interactions – even when non-significant, as this is more consistent with full reporting. In some places, non-significant p-values are summarized (e.g., all p>.33). I believe most of these are reported in full in tables, but please check, as I could not find full reporting of supplementary analyses.

Please report supplementary analyses in full. Results of key models are tabulated in text, along with mention of supplementary analyses in the text/ footnotes. If I understood correctly, the additional analyses are not reported/tabulated in full anywhere. The full analysis script is provided along with the data for analysis, but this requires a reader to run the analysis script to view the full statistical report. As the current software may become obsolete in the future, please add a supplementary analysis document to the OSF repository where the full statistical outputs can be viewed.

We have added tables for the supplementary analyses in Experiment 2. Those in Experiment 3 only have one predictor and so we didn’t think they needed full tables. 

Please clarify - is the 'simple' regression after removing random participant intercept still a repeated measures design? This is important because one person's responses may be correlated more closely to their own responses than to others' responses - even without the random effect being significant (i.e., people respond moderately differently to one another, but not significantly). By keeping the random factor in your model, you may actually improve the power to detect an effect. Perhaps there is different reasoning behind this decision, so I'd be happy to understand more about it - may be in a footnote or supplementary materials.

We now note in a footnote that the results (i.e., coefficient values and p values) are identical with or without the random effects. This is because there was zero variance in the random effects. 

Best practice in data visualization recommends that every figure summarizing the results of more than one participant include the participant N on the plot (e.g., in the legend) or in the Figure description, so that a reader doesn't have to go searching for the info in other parts of the article. Please add N to each graph or each age group. This also helps a reader to get an overview of whether the graph includes incomplete data across longitudinal samples (e.g., more samples at earlier than timepoints) or includes only those samples who were tested across all timepoint (i.e., no dropouts included in graph).

Have now added individual data points as well as N to all graphs.

Best practice in data viz is to include individual data points where possible, as well as an indication of confidence estimates about any point estimate drawn from more than one data point. Figure 3 uses bars to represent percentage of responses. This obfuscates the precise computation from 8 trials down to four bars: are we seeing the average value across the group for the individuals’ percentages in each condition (i.e., only possible values for each person are 100%, 50%, 0%)? Or something else? The authors could consider an alternative way of graphing to show individual binary responses – for example, dot plots - this would help to visually communicate the structure of the data, and make clearer that participants responses differ substantially across different items (e.g., almost at chance for ‘cheechee’)

Have now included box and whisker plots with individual participant data plotted.

Code for running the studies not yet provided in OSF. Please provide the scripts for the adult study (EPrime) and the eye-gaze studies (Habit). Including these would help make the study (or future adaptations) more replicable in the future.

Unfortunately this is not possible, as the testing computers used for the adult study are not accessible and, as far as we know, this is not possible for Habit. However, we have added the PsychoPy experiment file for Experiment 1b.

3D wireframes. Since the test objects were 3d printed, it would be great to include the wireframes used for 3D printing of the objects, or a link to a digital file on a maker-site. This would enhance future replicability of the project.

These have been uploaded to OSF.

Visual Stimuli For future replicability, please add photographs of the 3D shapes as used on screen in the study.

These have been uploaded to OSF. 

Audio Stimuli. One of the challenges of bouba/kiki studies internationally is knowing precisely what participants heard. One solution to this problem is accurate transcription in IPA, but as became clear in the metaanalysis by Styles & Gawne (2018), the notations provided in papers do not always align with strict IPA transcription. To allow greater transparency, please document the audio files in the OSF. As audio recording quality can differ substantially, including the audio files would readers to know what participants actually heard, including whether the content of the audio files can be easily heard by study participants. Archiving would also allow potential replicability into the future.

These have been uploaded to OSF.

Data files in archive have several different structures. It would be super helpful to include a code-book or a wiki explaining the different data files, their origin (what software & version?), and what each column represents in each file. Some of the Habit files share a structure, but for a casual viewer, it is important to know which columns are user-defined (e.g., Family and ID), and which of the automated columns are used in analysis. The Individual differences data file is particularly rich, but it is not clear where to look for the coding scheme if you start by looking at the data (the survey form I think?).

We have added variables descriptions on OSF.

Current archive only includes the survey from T1, but the text mentions that later surveys have different questions about babbling. Please include the full set of surveys for the different timepoints, with the precise questions answered by parents, and any protocol for implementation if additional questions were asked by a researcher.

• The archive appears to only include data from T3 – which appears in the report – but as data were collected at T1 and T2, it would be better to include a complete record of survey data at all three timepoints. Alternatively, if those data are reserved for another project, this should be described along with a link to where the data are archived separately. OSF allows archives to be created before being made public so if an embargo currently exists, a link can be provided for future accessibility.

• Many of the questions in the survey are not reported in the current writeup. There can be many reasons for not reporting all outcome measures, including that the bouba/kiki study reported here was part of a larger research project, which will be reported in full elsewhere. It is important to disclose the purpose for collecting tis data so a reader can know whether there was more exploratory analysis than the results reported here (i.e., multiple possible outcome measures were collected and only significant ones are reported in the current paper) or pre-conceived hypothesis tests. If exploratory, the analysis should be reported as such, along with the results of all non-significant tests, so that the probability values can be understood in context. If the study was preregistered, please clarify this in the main text of the study description so that the prereg can be readily identified.

We have uploaded a description of the developmental survey. The analysis is also now referred to as exploratory. We note that we only examined several of the variables that we judged to best operationalize potential developmental milestones. As to why we collected more responses than we analysed, the reason is simply that the survey was derived years before the analyses took place. We chose a plethora of variables to be sure we quantified any dimensions that could be of interest. When it came time to perform exploratory analyses, we only chose those that best captured the dimensions of interest. 

We have also uploaded a preliminary version of the survey data from Visits 1 and 2. Due to availability, we have only recently gotten access to these data. We will reupload a more “user friendly” version by June 30th.

Minor points included as comments in PDF. Context added by authors in square brackets.

The role of language specificity (i.e., phonotactics) is addressed briefly in the discussion, but perhaps could be made clearer from the perspective of possible learning trajectories.

Have added a note that the pattern differs by language. Then in the General Discussion now add that language as a “revelator” could be moderated by language, leading to differences in adult speakers of different languages. 

Also relevant: This procedure was non-blind to experimenter so children might be picking up other subtle paralinguistic or prosodic cues when identifying the expected response. [In reference to Maurer et al.]

Thank you for noting this however we elected not to include this information as, while an important point about the Maurer et al. study, it seemed tangential to the present study.

Not collected at the time of test? or not retained? 

Not retained. Have rephrased.

Please add a measure of confidence around this point estimate (e.g., CI?) or clarify [In reference to 71% result in 1a; same comment in 1b]

We have added these for Experiments 1a and 1b.

where can we find details of the analysis and how it was implemented? [In reference to follow up in 1a]

Have now added a table in the supplementary information.

This graph would be better if it were possible to view individual data points or distributions in the format of density plots [In reference to Fig 3]

Thank you for this suggestion, we have now plotted individual data points, along with box and whisker plots.

Table would be more informative if it included an indication of error/variance about the % listed - e.g., 95% Cis [Table 1]

We have added this.

Which written form? /bobo/ or "boeboe"? [Reference to 1b]

Have clarified.

More detailed description needed - same as the first analysis (with random factor for participants?) or the second analysis? [Reference to 1b result]

We have now added tables for supplementary analyses which include random factors. 

More detail needed (e.g., in supplement or code repository) [follow up in 1b]

Have now added a table in the supplementary information.

NB. Shang and Styles (2017) uses 3D shapes

Thank you for noting this. We have added a reference.

Great level of detail in demographics! Are the full data in the archive? [in reference to individual difference measures in E2]

These have not been included for the sake of privacy.

Not clear whether this developmental survey was intended for exploratory analysis of the results of the current study, or simply to characterize the kids in the study according to their general development. Alternatively, these data might have been collected for use in a different/larger study. Please clarify. [In reference to dev. Survey]

We now say that this was included “for the purposes of exploratory analyses to investigate if any developmental milestones predicted sensitivity to sound symbolism.”

Many of the questions in the survey are not reported in the current writeup. There can be many reasons for not reporting all outcome measures, including that the bouba/kiki study reported here was part of a larger research project, which will be reported in full elsewhere. It is important to disclose the purpose fro collecting tis data so a reader can know whether there was exploratory analysis (i.e., multiple possible outcome measures were collected and only significant ones are reported in the current paper) or pre-conceived hypothesis tests. If exploratory, the analysis should be reported as such, along with the results of all non-significant tests, so that the probability values can be understood in context. If the study was preregistered, please clarify this in the main text of the study description so that the prereg can be readily identified. [in ref to dev. Survey]

See above. 

We chose several variables for analyses that best captured our research questions and these are reported in text.

What is the degree of visual angle subtended by the two stimuli? this is important to understand as it may mean that children need to make just a saccade or a head movement to switch from one image to the other. This becomes important when considering the duration of looks included/discarded. [ref to E2]

The testing screen was large enough to allow infants to view both images without turning their head. The screen was 127cm on the diagonal. Infants were 150 cm away. This means that the screen subtended a visual angle of just over 45°. We have now added this information to the manuscript.

Is the code archived? [ref to Habit X 1.0 Software used to run Exp 2]

To the best of our knowledge it is not possible.

I'd like to hear more about your reasoning for this, and whether this was an a priori decision made data-blind (following an established protocol from previous study, before seeing current data), a condition-blind decision (applied after screening the data in a format where the condition of each trial was blind to the researcher making the decision), a results blind decision (made after a screening process where the condition of each trial was known, but before any between condition statistical comparison had been conducted), or a post-hoc decision after a preliminary analysis was conducted. [Reference to excluding look times < 2 sec]

We provided more details regarding the 2 second look time. See below:

We set a minimum look time of 2 seconds so that the infant had the opportunity to hear at least one token of the word. On the rare occasion that an infant did not attend for a minimum of two seconds, the bouncing ball and pre-test were repeated to recapture the infant’s attention. After the test trials, we presented a post-test, which was identical to the pre-test (i.e., a spinning pinwheel with music for 20 seconds).

Please add effect size for all effects including n.s. ones as this allows for future metaanalyses 

Thank you for this suggestion. We have now included tables for all supplementary analyses in online supplementary material. However, if you meant for a measure of effect size beyond regression coefficients, we would be happy to include those!

Please include sample size on figure, in Title or Note 

Thank you for this suggestion, they have been added to the notes under all figures.

Please unpack this a little for the reader - what does this mean for interpretation of the null hypothesis? 

We have provided a more easily interpreted wording for this: “In other words, this analysis asks: when taking a range of values that we are 95% sure contain the true value of a coefficient, what percentage are practically equivalent to 0.”

unclear reasoning here... I understand that there were different surveys at different time points, but am I correct that only visit 3 is included in this section of the paper? Both for the survey and for the test data? It would be great to hear more about the reasoning for only testing T3.

We’ve now expanded on our reasoning for analysing individual difference scores from visit three (and hopefully clarify that only those individual difference measures were analysed): 

Our goal was to investigate if any individual difference measures were related to the emergence of sensitivity to sound symbolism. Because there was only a trend towards sensitivity to sound symbolism developing at visit three, we only analyzed individual difference measures from that visit. We then analysed whether these were related to sound symbolism scores from that visit.

Perhaps I wasn't following clearly, but I wasn't sure how time was implemented for each of these correlations. The surveys were conducted at 3 time points, as were the gaze direction tests. Were the factor analyses run separately for each time point and then correlations were tested only for synchronous correspondence between survey and test? Or were all score across time points collapsed into a single measure of individual difference? 

Hopefully the above clarification also helps with this.

This text is unclear. Perhaps better to split into two tables, or two columns in one table. One for table/column could be for binary milestones, where the number reported is % of children, and one table/column where the number is the mean score.

Thanks for this suggestion, we have now separated the two types of measures.

Might be nice to present these data together with some indication of error/variance e.g., Confidence Intervals [In reference to table 4]

These are the percentage of children in the sample demonstrating each particular skill, so we aren’t sure if that would be appropriate?

Please unpack this a bit more - was this a mistake by the parent? by the experimenter? or something else? [regarding E3 note about mistake in label pronunciation]

This was based on an incorrect pronunciation by the caregiver when prompted to produce the label during the second visit. This detail has been added to the manuscript.

Was this based on a a priori power analysis or some other rubric? [sample size in E3]

This was based on comparable studies in the literature (see response to Experiment 2) as well as practical limitations given the demanding nature of the study. We now say as much in the manuscript. In addition, this sample size was the size that we included in the original grant proposal for this project. 

This analysis seems different from the previous where the analysis was conducted in terms of shape first and label second. Did I miss something? [E3 results]

It is indeed different. We used looking time to congruent object instead of round object to avoid having a threeway interaction (i.e., between nonword label, visit, and training congruent). 

Need some data display to contextualize the results relative to the previous study (even if non significant [E3 results] 

We have now added a figure.

See also Shang & Styles (2017, 2023)for differences in the nature of sound symbolism for langs with different acoustic features. 

Thank you for pointing us to these interesting references, however we have elected not to include them here as we read them as being on a slightly different topic.

May be better to rephrase as with optimistic framing - e.g., although it would not survive correction for multiple comparisons in the current study, it suggests a new target for hypothesis testing in future work.

Have rephrased in this way.

---

## [Editor Report · Decision Letter 1]

14 Jun 2023

The Maluma/Takete Effect is Late: No Longitudinal Evidence for Shape Sound Symbolism in the First Year

PONE-D-22-33843R1

Dear Dr. Sidhu,

We’re pleased to inform you that your manuscript has been judged scientifically suitable for publication and will be formally accepted for publication once it meets all outstanding technical requirements.

Kind regards,

Marcus Perlman, Ph.D

Academic Editor

PLOS ONE
---

## [Editor Report · Acceptance letter]

26 Jun 2023

PONE-D-22-33843R1 

The maluma/takete effect is late:
no longitudinal evidence for shape sound symbolism in the first year 

Dear Dr. Sidhu:

I'm pleased to inform you that your manuscript has been deemed suitable for publication in PLOS ONE. Congratulations! Your manuscript is now with our production department. 

Kind regards, 

on behalf of

Dr. Marcus Perlman 

Academic Editor

PLOS ONE